# SEPARATE: A Simple Low-rank Projection for Gradient Compression in Modern Large-scale Model Training Process

**Hanzhen Zhao**[1], **Xingyu Xie**[2], **Cong Fang**[1,3,*], **Zhouchen Lin**[1,3,4]

{hzzhao, xyxie, fangcong, zlin}@pku.edu.cn

[1] State Key Lab of General AI, School of Intelligence Science and Technology, Peking University
[2] Department of Mathematics, National University of Singapore
[3] Institute for Artificial Intelligence, Peking University
[4] Pazhou Laboratory (Huangpu), Guangzhou, Guangdong, China

## Abstract

Training Large Language Models (LLMs) presents a significant communication bottleneck, predominantly due to the growing scale of the gradient to communicate across multi-device clusters. However, how to mitigate communication overhead in practice remains a formidable challenge due to the weakness of the methodology of the existing compression methods, especially the neglect of the characteristics of the gradient. In this paper, we consider and demonstrate the low-rank properties of gradient and Hessian observed in LLMs training dynamic, and take advantage of such natural properties to design SEPARATE, a simple low-rank projection for gradient compression in modern large-scale model training processes. SEPARATE realizes dimensional reduction by common random Gaussian variables and an improved moving average error-feedback technique. We theoretically demonstrate that SEPARATE-based optimizers maintain the original convergence rate for SGD and Adam-Type optimizers for general non-convex objectives. Experimental results show that SEPARATE accelerates training speed by up to 2× for GPT-2-Medium pre-training, and improves performance on various benchmarks for LLAMA2-7B fine-tuning.

## 1 Introduction

With massive amounts of data, billions of parameters, and multi-device clusters, the remarkable strides of Large Language Models (LLMs) across multiple disciplines [9, 48, 5, 8] are attributed to such scalable intrinsic characteristics and training paradigms. However, corresponding challenges erupt, as multiple rounds of communication of massive model parameters across multi-device clusters have created a significant communication bottleneck in training process. For example, pre-training a BERT-Large model (340M parameters) with a single batch size on 16 GPUs shows that 92% of the training time is spent on gradient all-reduce in backward propagation [49]. Therefore, a critical problem urgently to be addressed is how to relieve the communication burden while ensuring the quality of model training.

In response to the aforementioned problem, a variety of gradient compression strategies have been proposed to reduce the communication overhead of gradients between devices. Such strategies can be broadly categorized into two compression techniques, including low-precision compressors (e.g., SignSGD [6], 1-bit Adam [49] and 1-bit LAMB [29]) and low-rank compressors (e.g., Atomo [57] and PowerSGD [55]). Moreover, error-feedback technique [46, 49, 29] focuses on compensating for errors accumulated during compression by changing the objects to compress from the gradient to the gradient and historical error summation [49, 29]. Error-feedback-based compression algorithms have been theoretically proven to have lower communication complexity when applied to SGD-Type optimizers [53, 30, 18].

---

*Corresponding author

However, from the system perspective, how to mitigate communication overhead in practice remains a core problem and a formidable challenge. For example, low-bit gradients (e.g., less than 8-bit) are not supported by typical hardware. Thus, quantizing gradients into less than 8 bits can lead to significant precision loss and numerical instability. For biased low-precision compressors with error-feedback techniques, which do not support communication primitive all-reduce due to their inherent structures, they must use all-gather for aggregation. It limits them only suitable for master-server communication patterns, incompatible with advanced ring-based and tree-based patterns in current large-scale model training [38] and extremely slows down the communication speed [1]. Therefore, there remains a significant gap between the performance of these methods in training practice and their theoretical results. Considering the system-level speed, recent studies [1, 62] indicate that when training representative LLMs with off-the-shelf DistributedDataParallel (DDP), most of gradient compressors show longer wall-clock training time than vanilla Adam. Furthermore, integrating these gradient compressors into commonly used optimizers often requires significant modifications, necessitating additional effort to ensure compatibility and maintain effective.

These system-level limitations drive us to consider improvements at the methodological level. The ineffectiveness of these methods stems from inherent flaws in their methodology. Specifically, these algorithms seldom take the characteristics of the gradient of LLMs into account, leading to additional computation or communication rounds to compensate for the errors introduced by compression. For low-precision compressors, the compression ratio is upper bounded by floating-point digit number 32. Worse, vector-wise quantization is independent of the properties of the gradient and is commonly computationally heavy. Several low-rank compressors [55] consider the low-rank approximation of gradient, but the compression and decompression are so complex that the time cost in extra computation is close to or even larger than the saved communication time cost. Therefore, the design concept of modern "workable" gradient compression algorithms is that

**We should design compression algorithms by leveraging the properties of the model, ensuring the suitability for modern LLMs efficient training ecology, and lightweighting compression and decompression computation to save time.**

**Contribution.** In this work, we focus on how to design a universal gradient compression technique in line with the above concept. We propose SEPARATE, a SimplE low-rank Projection for grAdient compRession in modern lArge-scale model Training procEss, which compresses the gradient to arbitrary low-dimension one before communication and then reconstructs after, no matter what optimizers and training frameworks are used. Instead of a low-rank approximation estimate of weight matrix for parameter-efficient fine-tuning like LoRA [22], we stand on the low-rank structure of the gradient itself.

The motivation of SEPARATE originally comes from the observation of geometric properties of the Hessian spectrum during training dynamic. Especially, many studies reveal that the eigen-spectrum of Hessian is often "top-heavy" [55, 43, 44, 61], indicating that several top eigenvalues are dominant the trace of Hessian. This fact intuitively suggests the theoretical feasibility of low-rank gradient compression with restricted variance through careful designed compression strategy. We show the existence of such low-rank properties in Section 3, and the theoretical validity of SEPARATE in Section 5.

Let us give a general introduction to SEPARATE to illustrate its effectiveness. Considering gradient $\mathbf{g} \in \mathbb{R}^d$ to communicate, we generate a common Gaussian random matrix $\mathbf{M} \in \mathbb{R}^{d \times m}$ on each node and do $\mathbf{p} = \mathbf{g} \times \mathbf{M}$ to compress $\mathbf{g}$. After communication, we use the same random matrix for decompression $\tilde{\mathbf{g}} = \mathbf{p} \times \mathbf{M}^\top$. Considering that $\mathbb{E}[\frac{1}{m}\mathbf{p}\mathbf{M}\mathbf{M}^\top] = \mathbf{p}\mathbb{E}[\frac{1}{m}\mathbf{M}\mathbf{M}^\top] = \mathbf{p}$, we show the compressor is unbiased, and the compression ratio $m$ is arbitrary. Moreover, the variance analysis seems to be more important. We show theoretically that the variance is bounded in Section 4, and the gradient complexity for SGD and Adam-Type optimizers maintains the same order as vanilla SGD and Adam in Section 5.

We demonstrate that SEPARATE works well in both LLMs pre-training and fine-tuning tasks. To reduce compression error in training process, we design a novel error-feedback mechanism for SEPARATE. We estimate more stable compression errors using a moving average of historical errors and incorporate these errors back into the gradient before compression. We also reset the error periodically to eliminate bad historical information and select update directions preciously. For

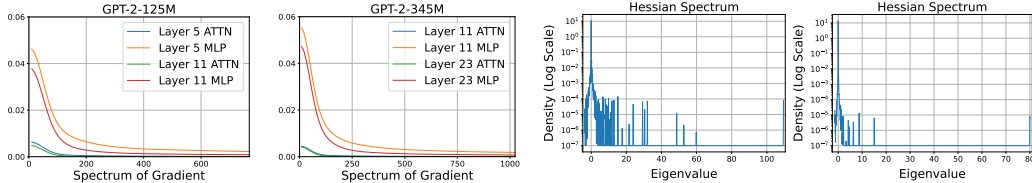

Figure 1: Low-rank gradient and Hessian observations in practice. The two pictures on the left show the spectrum of gradient matrices of chosen layers in GPT-2-125M and GPT-2-345M respectively. The X-axis represents the order of eigenvalues from largest to smallest, and the Y-axis represents the magnitude of eigenvalues. The two pictures on the right show the spectrum of Hessian in ResNet-20 and ResNet-110 trained on CIFAR-10 respectively. The X-axis represents the eigenvalue, and the Y-axis represents the corresponding density.

GPT-2 [40] pre-training, SEPARATE can improve the training speed of AdamW by $2\times$ times. For LLAMA2 [51] fine-tuning, SEPARATE can obtain better performance on downstream tasks.

As a gradient projection method, SEPARATE is independent of the choice of optimizers and can be regarded as a plug-in, which can be simply implemented with a few lines of code in off-the-shelf frameworks as shown in Algorithm 2. In addition, the method is robust to some extent, and its performance is insensitive to the selection of hyperparameters. As long as the selection of compression ratio meets the random direction sampling quantity required in theoretical analysis, the algorithm can show effective performance.

## 2 RELATED WORK

**Communication Efficient Training.** As the scale of models explosively grows in recent years [48, 36, 25, 52], the redundant communication of gradient has become the main bottleneck of training. Gradient compression [29, 55, 49, 46, 56, 34] is a promising solution for communication-efficient training among GPUs clusters. In general, the compression techniques can be divided into low-precision methods [49, 29, 46, 34] and low-rank methods [55, 32, 2]. Besides, several theoretical and empirical analyses of more general unbiased compressor $\mathcal{C}$ [18, 53, 21] are also proposed to show the effect of gradient compression theoretically. Moreover, ZeRO++ [56] has been proposed to apply quantization techniques to sharding strategy [41].

**Error-feedback Technique.** Gradient compression techniques bring information loss during the training process and impair the training accuracy and performance of the model. The accumulation of error may cause algorithm divergence [46]. To solve such a problem, the error-feedback technique was first proposed by [46], which adds the compression error to the gradient before compression in this iteration. Afterward, various works study the effect of the error feedback technique from theory to practical methods. Recent works [18, 53] study the theoretical convergence of unbiased compressor $\mathcal{C}$ with error-feedback technique on convex and non-convex settings. Moreover, many induce the error-feedback technique to adaptive gradient algorithms[27, 15] which is commonly used in training of large models, such as 1-bit Adam [49] and 0/1 Adam [34]. LoCo [60] first presents the error-feedback-based efficient training method for the sharding strategy.

**Low-rank Approximation.** Many studies focus on the low-rank approximation of weight or gradient matrix in neural network training dynamic and propose the error bound [55, 22]. However, such a low-rank approximation of weight (e.g., LoRA [22]) may not reach a comparable performance as full-rank in pre-training [31] and fine-tuning [58]. Some studies show that the Hessian in deep neural networks is "top-heavy" [43, 33, 44], which means the eigenvalue drops fast and the Hessian maintains low-rank, and some recent studies theoretically illustrate that the gradient is naturally low-rank [13, 63]. In fact, some classical methods have considered low-rank estimation of trace, like the Hutchinson estimator [23]. Such related works provide the potential to design a simple gradient compression by low-rank projection with restricted variance. Some recent research about training or gradient accumulation consider the low-rank property of gradient or Hessian, and use random projection for memory reduction [67, 19], but lack deep analysis on bounded variance.

---

**Algorithm 1** SEPARATE: A Simple Low-rank Projection for Gradient Compression

---

**Require:** Initialization model parameters, $N$ nodes, one-round communication budget $m$, error reset frequency $T$, $\beta \in [0, 1]$, a common Gaussian random number generator, initialize $\mathbf{e}^0 \in \mathcal{B}(\mathbf{0}, c_1)$

   **while** $k \leq K$ **do**

      STEP 1. In each node $n$ compute stochastic gradient $\mathbf{g}_n^k$ and $\mathbf{h}_n^k = \mathbf{g}_n^k + \mathbf{e}_n^k$;

      STEP 2. Generate fresh i.i.d. common random Gaussian vectors $\boldsymbol{\xi}_1, \cdots, \boldsymbol{\xi}_m \sim N(0, \mathbf{I}_d)$ and compute $[p_{1,n}, \cdots, p_{m,n}]$ with $p_{i,n} = \langle \mathbf{h}_n^k, \boldsymbol{\xi}_i \rangle$ as the low-dimension projection of $\mathbf{h}_n^k$;

      STEP 3. Do all-reduce and obtain global projected gradient $[\tilde{p}_1, \cdots, \tilde{p}_m]$;

      STEP 4. Compute $\tilde{\mathbf{h}}_n^k = \frac{1}{m} \sum_{i=1}^m \tilde{p}_i \cdot \boldsymbol{\xi}_i$ and use $\tilde{\mathbf{h}}_n^k$ for model weight update in node $n$;

      STEP 5. Update error $\mathbf{e}_n^{k+1} = (1 - \beta)\mathbf{e}_n^k + \beta(\mathbf{h}_n^k - \tilde{\mathbf{h}}_n^k)$ if $k\%T \neq 0$, or $\mathbf{e}_n^{k+1} = 0$ if $k\%T = 0$ (error reset);

   **end while**

---

# 3 Low-rank Property of Gradient and Hessian in Training

Through some phenomena observed in practice and simple theoretical analysis of Transformer [54], we further illustrate the widespread existence of the low-rank property of gradient. Moreover, we also demonstrate that the eigen-spectrum of Hessian is "top-heavy", consistent with some previous work about Hessian spectrum of neural networks [43, 44]. We study these observations practically in this work. As shown in Figure 1, the landscape of gradient in representative LLMs training possesses low-rank property. Specifically, We observe the spectrum of GPT-2-Small (125M, with 12 Transformer-based layers) and GPT-2-Medium (345M, with 24 Transformer-based layers). We observe the gradient spectrums of attention and MLP in middle layer and last layer, respectively. The eigenvalues of grad drop fast, especially in **MLP** layers. This phenomenon is more pronounced in larger models. Based on recent study on Transformer's dynamic [50], we theoretically prove that the gradient of MLP layer in Transformer shows low-rank property over time in Appendix D.

Moreover, we study the spectrum of Hessian of trained deep neural networks. In fact, early studies [43]have discovered that the eigenvalues of simple networks (e.g., 3-layer networks) drop fast. As shown in Figure 1, we study the spectrums of Hessian of different scales of ResNet [20] trained on CIFAR-10 [3]. Except several extremely large eigenvalues, most of eigenvalues of the Hessian are close to zero. It is more obvious in larger network structures. This means that a lot of the directions along the loss are almost flat, and the trace of Hessian is bounded much less than $\mathcal{O}(d)$, which ensures our theoretically analysis in Section 5.

# 4 SEPARATE: A Simple Low-rank projection

Considering low-rank property of gradient and "top-heavy" observation of Hessian, a natural idea of gradient compression is to randomly project the gradient to low-dimensional subspace. As the sample is up to a certain level (usually much less than dimension $d$), the dominant information of gradient can be exactly recovered. This leads to our SEPARATE method. The core idea of SEPARATE is composed with two parts. First, we design a Gaussian random projection compressor for dimensionality reduction of gradient. Hessian-trace-bounded proposes the theoretical guarantee for random projection. Second, to solve the critical challenge that the compression error in gradient compression would accumulate during the training process, we propose a novel moving average error feedback technique to ensure practical utility. We introduce SEPARATE method in Algorithm 1.

## 4.1 Common Random Projection Compressor

For classic multi-node parallel training, each node computes its local stochastic gradient $\mathbf{g}_n^k$ with their mini-batch data. Then through communication like all-reduce each node obtains the global gradient for model update.

As mentioned early, we take the low-rank property of gradient and "top-heavy" property of Hessian into account to design a variance-bounded low-rank projection for gradient compression, applicable to most of practical scenarios in large-scale model training. Specifically, for the gradient vector $\mathbf{g}$, each node generates $m$ common random Gaussian variables $\boldsymbol{\xi}_i, \cdots, \boldsymbol{\xi}_m \sim N(0, \mathbf{I}_d)$. Then each node

computes

$$p_i = \langle \mathbf{g}, \boldsymbol{\xi}_i \rangle, \forall i \in [m]. \tag{1}$$

After compression, each node constructs $[p_1, \cdots, p_m]$ as the compressed gradient for communication like all-reduce. Then they obtain the global $[\tilde{p}_1, \cdots, \tilde{p}_m]$, and reconstruct the gradient as

$$\tilde{\mathbf{g}} = \frac{1}{m} \sum_{i=1}^{m} \tilde{p}_i \cdot \boldsymbol{\xi}_i. \tag{2}$$

Then we theoretically illustrate the unbiasedness and variance-bounded property of our common random projection in Lemma 4.1 and Lemma 4.2 as below. The proof details are shown in Appendix B.

**Lemma 4.1.** $\tilde{\mathbf{g}}$ *is an unbiased estimator of* $\mathbf{g}$,

$$\mathbb{E}_{\boldsymbol{\xi}_1, \cdots \boldsymbol{\xi}_m} \tilde{\mathbf{g}} = \mathbf{g}. \tag{3}$$

**Lemma 4.2.** *The variance of* $\tilde{\mathbf{g}}$ *under norm* $\|\cdot\|_{\mathbf{A}}$*, where* $\mathbf{A}$ *is a given positive semi-definite symmetric matrix, can be bounded by* $\frac{3\mathrm{tr}(\mathbf{A})}{m}\|\mathbf{g}\|^2 - \frac{1}{m}\|\mathbf{g}\|_{\mathbf{A}}^2$,

$$\mathbb{E}_{\boldsymbol{\xi}_1, \cdots, \boldsymbol{\xi}_m} \|\tilde{\mathbf{g}} - \mathbf{g}\|_{\mathbf{A}}^2 \leq \frac{3\mathrm{tr}(\mathbf{A})}{m}\|\mathbf{g}\|^2 - \frac{1}{m}\|\mathbf{g}\|_{\mathbf{A}}^2. \tag{4}$$

**Remark 4.3.** *Lemma 4.1 directly indicates the unbiasedness of common random projection, which is standard for low-rank compression techniques [55, 67, 19]. What we really highlight is Lemma 4.2, which illustrate that the variance of our compression strategy under the Mahalanobis norm can be bounded by the trace of given matrix. Considering the standard convergence analysis of SGD or Adam-Type optimizers, the second order factor of Taylor expansion of the objective function can be written as Mahalanobis norm under Hessian matrix, and the "top-heavy" property ensures the trace of Hessian is much smaller than* $dL$*, where* $d$ *is the dimension and* $L$ *is Lipschitz constant defined in Assumption 5.2. This provides the potential for SEPARATE-based algorithms to surpass other compressors in the order of* $d$ *in convergence speed. We show the theoretical results in Section 5.*

### 4.2 MOVING AVERAGE ERROR FEEDBACK

Error-feedback technique is widely used in gradient compression methods [46, 42, 60] to mitigate the information loss during compression. It is worth considering how error-feedback works for our common random projection compressor, even though Lemma 4.1 and Lemma 4.2 show the unbiasedness and trace-bounded variance, because the accumulated error in continuous updates may cause wide deviation. One straight solution is to use the gradient compression error from the previous iteration as $\mathbf{e}_n^k = \arg\min_{\mathbf{e} \in \mathbb{R}^d} \left\| \mathbf{e} - \left( \tilde{\mathbf{h}}_n^k - \mathbf{g}_n^k \right) \right\|^2 = \tilde{\mathbf{h}}_n^k - \mathbf{g}_n^k$, where $\mathbf{g}_n^k$ is the original gradient on the $n$-th node and $\tilde{\mathbf{h}}_n^k$ is the reconstruction of compressed gradient. However, we empirically discover the instability of this estimate, because for single-step iteration, random vectors for projection may have a large deviation from the direction of true gradient. This means potential abrupt fluctuations of $\mathbf{e}_n^k$, especially for several continuous iterations with a series of random vectors with very different directions. This may lead to the whole training process converging to another suboptimal region. This phenomenon is particularly prominent when training from scratch, changing the shape of loss curve and the convergence result of loss (see Section 6.3). To solve this problem, inspired by the idea of momentum technique in the analyses of accelerated gradient descent [37], we take the moving average of the historical compression error and the current one to maintain the continuity to some degree:

$$\mathbf{e}_n^k = \underset{\mathbf{e} \in \mathbb{R}^d}{\arg\min} \frac{\beta}{2} \left\| \mathbf{e} - \left( \tilde{\mathbf{h}}_n^k - \mathbf{g}_n^k \right) \right\|^2 + \frac{1-\beta}{2} \left\| \mathbf{e} - \mathbf{e}_n^{k-1} \right\|^2 = (1-\beta)\,\mathbf{e}_n^{k-1} + \beta \left( \tilde{\mathbf{h}}_n^k - \mathbf{g}_n^k \right), \tag{5}$$

where $\beta \in [0, 1]$ represents the trade-off between two factors. We demonstrate that the moving average error feedback can reduce the total accumulated error when applying it to SGD or Adam-Type optimizers in Lemma C.2 and Lemma C.3. In brief, we simply formulate the iteration of SGD and Adam-Type optimizers as $\boldsymbol{\theta}^k = \boldsymbol{\theta}^0 - \sum_{i=i}^{k} \boldsymbol{\eta}^i \circ \mathbf{g}^i$, where $\boldsymbol{\theta}$ means the model's weight and $\circ$ means

the Hadamard product of two vectors. Our analysis shows that moving average error-feedback effectively reduces the accumulated gap of weights updated by $\mathbf{g}$ and by $\tilde{\mathbf{g}}$ as

$$\left\| \sum_{i=1}^{k} \boldsymbol{\eta}^i \circ \mathbf{g}^i - \sum_{i=1}^{k} \boldsymbol{\eta}^i \circ \tilde{\mathbf{g}}^i \right\| \leq \mathcal{O}\left( \eta \|\mathbf{e}^k\| \right), \tag{6}$$

which means as the training goes on, the accumulated compression error of the top $k$ rounds is only of $\mathbf{e}^k$-order, instead of the accumulation from $\mathbf{e}^0$ to $\mathbf{e}^k$. Compared with vanilla error feedback technique [46], taking moving average of the historical error maintains stability to some extent and reduces the variance of accumulated error.

Moreover, we consider the compression error in the early training process cannot guide the later iteration, so we reset the compression error after a given number of iterations like 128. Combining the techniques above we propose SEPARATE method in Algorithm 1. Our SEPARATE can be applied to all the gradient-based optimizers with a few lines of codes, as shown in Algorithm 2.

## 5 CONVERGENCE GUARANTEE

In this section, we propose the convergence guarantee of our SEPARATE method. We consider the following non-convex optimization problem

$$\min_{\boldsymbol{\theta}} f(\boldsymbol{\theta}) = \mathbb{E}_{\zeta \sim \mathcal{D}} \left[ F(\boldsymbol{\theta}, \zeta) \right], \tag{7}$$

where $F$ is the non-convex optimization objective and $\boldsymbol{\theta}$ is the model weight to update. Data is sampled from a given distribution $\mathcal{D}$. We focus on the convergence rate of SEPARATE-based SGD and Adam-Type Optimizer to find an $\epsilon$-approximate first-order stationary point of the objective $f(\cdot)$. For such SEPARATE-based optimizers, they update the model weight as below:

$$\mathbf{SGD}: \boldsymbol{\theta}^{k+1} = \boldsymbol{\theta}^k - \eta\tilde{\mathbf{h}}^k, \qquad \mathbf{Adam\text{-}Type}: \begin{cases} \mathbf{m}^k = (1 - \beta_1)\mathbf{m}^{k-1} + \beta_1\tilde{\mathbf{h}}^k, \\ \boldsymbol{\eta}^k = \eta \times v\left( \tilde{\mathbf{h}}^0, \cdots, \tilde{\mathbf{h}}^k \right), \\ \boldsymbol{\theta}^{k+1} = \boldsymbol{\theta}^k - \boldsymbol{\eta}^k \circ \mathbf{m}^k, \end{cases} \tag{8}$$

where $\beta_1 \in (0, 1)$, and $v(\cdot)$ computes a series of pre-conditioners based on different AdamType optimizers. For example, $v(\cdot)$ computes the inverse of the second order moment of Adam's gradient like $v\left( \tilde{\mathbf{h}}^0, \cdots, \tilde{\mathbf{h}}^k \right) = 1/\sqrt{\mathbf{v}^k + \delta}$ where $\mathbf{v}^k = (1 - \beta_2)\mathbf{v}^{k-1} + \beta_2(\tilde{\mathbf{h}}^k)^2$. Adam-Type optimizers differ from various definitions of $v(\cdot)$ [27, 15, 47]. We analyze them uniformly.

Like the common analysis of the convergence rate of general non-convex objective [24, 59], we define the $\epsilon$-approximate first-order stationary point as below.

**Definition 5.1** ($\epsilon$-stationary point). *The model weight $\boldsymbol{\theta}$ is an $\epsilon$-appriximate first-order stationary point of $f$ if $\|\nabla f(\mathbf{x})\| \leq \epsilon$.*

Definition 5.1 means for SGD and Adam-Type optimizers, we need to find the iteration round $K$ such that $\min_{k \in [K]} \mathbb{E} \left\| \nabla f(\boldsymbol{\theta}^k) \right\| \leq \epsilon$, or more tightly

$$\frac{1}{K} \sum_{k=0}^{K-1} \mathbb{E} \left\| \nabla f(\boldsymbol{\theta}^k) \right\|^2 \leq \epsilon^2. \tag{9}$$

Moreover, we formally present some assumptions to constrain the objective function and the optimization problem. Assumption 5.2 and 5.3 are commonly used to describe the properties of the objective in stochastic setting [24, 16]. Assumption 5.4 ensures the trace of Hessian is bounded globally, which is also common in practical applications [43].

**Assumption 5.2** (L-smoothness). *The function $f$ is L-smooth if it satisfies*

$$\|f(\boldsymbol{\theta}_1) - f(\boldsymbol{\theta}_2)\| \leq L\|\boldsymbol{\theta}_1 - \boldsymbol{\theta}_2\|, \quad \forall \boldsymbol{\theta}_1, \boldsymbol{\theta}_2 \in \mathbb{R}^d. \tag{10}$$

**Assumption 5.3** (Stochastic Gradient Boundness). *The stochastic gradient $\mathbf{g}^k$ on each device is unbiased and its infinite norm and variance are bounded as:*

$$\mathbb{E}\|\mathbf{g}^k\|_\infty \leq c_\infty, \qquad \mathbb{E} \left\| \nabla f(\boldsymbol{\theta}^k) - \mathbf{g}^k \right\|^2 \leq \sigma^2, \quad \forall k \in [K]. \tag{11}$$

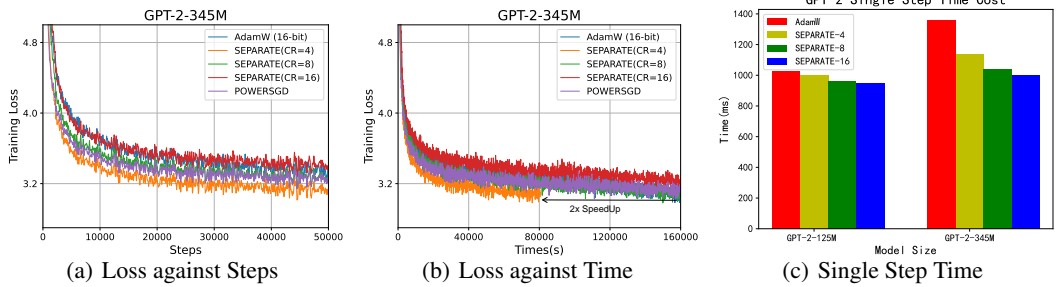

Figure 2: Loss curve and single step time cost of vanilla AdamW and SEPARATE with different compression ratios (CR) on GPT-2-345M trained with 10B tokens OpenWebtext dataset. (a) shows the loss against the iteration steps, and (b) shows the loss against the wall-clock time. (c) shows the single-step time cost of vanilla AdamW and SEPARATE with different compression ratios.

**Assumption 5.4** (Hessian-domination). *The function $f$ is $\mathbf{A}$-Hessian dominated if there exists a positive semi-definite symmetric matrix $\mathbf{A}$ such that $\nabla^2 f(\boldsymbol{\theta}) \preceq \mathbf{A}, \forall \boldsymbol{\theta} \in \mathbb{R}^d$.*

Now we present the main theorem of the convergence rate of SEPARATE-based SGD and Adam-Type optimizers in general non-convex setting. The proof of two Theorems can be seen in Appendix C.

**Theorem 5.5** (SEPARATE-based SGD convergence). *Suppose Assumption 5.2, 5.3 and 5.4 hold. Let $\mathbf{e}^0 \in \mathcal{B}(\mathbf{0}, c_1)$, namely $\forall \mathbf{x} \in \mathcal{B}(\mathbf{0}, c_1), \|\mathbf{x}\| \leq c_1$. Let $m \leq \frac{\mathrm{tr}(\mathbf{A})}{L}$ and $\eta \leq \frac{m}{4\mathrm{tr}(\mathbf{A})}$. With $\eta = \mathcal{O}\left(d^{-1/2}\epsilon^2\right)$ in SEPARATE-based SGD, after $K = \Omega\left(d^{1/2}\epsilon^{-4}\right)$ iterations in Algorithm 1, it holds*

$$\frac{1}{K} \sum_{k=0}^{K} \mathbb{E} \left\| \nabla f(\tilde{\boldsymbol{\theta}}^k) \right\|^2 \leq \mathcal{O}\left(\epsilon^2\right). \tag{12}$$

**Remark 5.6.** *This improved convergence rate under standard non-convex stochastic setting is derived from our analysis of trace-bounded variance in Lemma 4.2. In the common analysis of compression techniques, compression reduces the amount of information through single step communication by $\mathcal{O}(d)$ times, or with $\mathcal{O}(d)$-variance [2]. Moreover, considering error feedback technique brings bias, the analysis is commonly element-wise [2, 46]. Thus the convergence rate is often dimensional dependent $\mathcal{O}\left(d\epsilon^{-4}\right)$, but they assume it is default and usually ignore this item [46]. Considering modern LLMs training, the dimension of parameters is often extremely large and cannot be ignored. Fortunately, our analysis provides an improved rate on the order of $d$ from $\mathcal{O}(d)$ to $\mathcal{O}\left(d^{1/2}\right)$ under natural defined Assumption 5.3 from "top-heavy" Hessian observation. It is the first gradient compression method to achieve such rate to the best of our knowledge.*

**Theorem 5.7** (SEPARATE-based Adam-Type convergence). *Suppose Assumption 5.2, 5.3 hold. Assume $c_l \leq \|v(\cdot)\|_\infty \leq c_u$ and $\|\boldsymbol{\eta}^k - \boldsymbol{\eta}^{k-1}\|_\infty \leq \eta\beta_1(1-\beta_1)^{K-k}c_u$. Let $\mathbf{e}^0 \in \mathcal{B}(\mathbf{0}, c_1)$, $\eta \leq \min\left\{\frac{c_l}{2Lc_u^2}, \frac{\beta_1 c_l^{0.5}}{5c_u^{1.5}(1-\beta_1)L}\right\}$. With $\eta = \mathcal{O}\left(d^{-1}\epsilon^2\right)$ and $\beta_1 = \mathcal{O}\left(d^{-1}\epsilon^2\right)$ in SEPARATE-based Adam-Type optimizers, after $K = \Omega\left(d\epsilon^{-4}\right)$ iterations in Algorithm 1, it holds*

$$\frac{1}{K} \sum_{k=0}^{K-1} \mathbb{E} \left[ \left\| \nabla f(\tilde{\boldsymbol{\theta}}^k) \right\|^2 + \frac{1}{4} \left\| \mathbf{m}^k \right\|^2 \right] \leq \mathcal{O}\left(\epsilon^2\right). \tag{13}$$

**Remark 5.8.** *Because different coordinates of the gradient are updated with different step sizes in Adam-Type optimizers, the convergence rate is inevitably dependent on dimension $d$, and combined with compression and error feedback, Adam-Type optimizers are often $\mathcal{O}(d)$ at the order of $d$. Theorem 5.7 shows that integrating SEPARATE with various Adam-Type optimizers does not affect their convergence speed. Thus SEPARATE-based Adam-Type optimizers share the same stochastic gradient complexity as $\mathcal{O}\left(\epsilon^{-4}\right)$ when ignoring the dimensional factor $d$.*

Table 1: Performance comparison with representative communicatio-efficient methods. We fine-tune LLAMA2-7B on the alpaca-gpt4 dataset and evaluate the performance on downstream tasks, including commonsense reasoning, world knowledge, math, and code.

| Method | TriQA | GSM8K | MBPP | NQ | WinoG | Arc-e | Arc-c | PIQA | HellaS | Avg.S | Avg.R |
|---|---|---|---|---|---|---|---|---|---|---|---|
| Adam | 49.39 | 15.69 | 19.40 | 3.07 | 46.09 | 72.31 | **53.90** | **57.07** | 26.45 | 38.15 | 3.22 |
| PowerSGD | 50.78 | 18.65 | **22.40** | 2.52 | 40.57 | 62.43 | 42.71 | 52.88 | 30.12 | 35.90 | 3.00 |
| 1-bit Adam | **62.08** | 16.53 | 16.80 | **5.15** | 49.57 | 49.21 | 37.97 | 52.88 | 22.04 | 34.69 | 3.22 |
| ZeRO++ | 57.49 | 18.42 | 22.20 | 1.72 | **49.88** | 43.03 | 30.85 | 47.61 | **33.97** | 33.91 | 3.22 |
| SEPARATE | 57.18 | **20.17** | 21.40 | 4.18 | 49.25 | **72.66** | 49.83 | 52.99 | 29.73 | **39.71** | **2.22** |

# 6 EXPERIMENTS

## 6.1 SOTA COMPARISON WITH COMMUNICATION EFFICIENT METHODS

We evaluate SEPARATE on both pre-training and fine-tuning tasks of LLMs. The experiment setting details are shown in Appendix E.

**Pre-Training GPT-2 from Scratch.** We train GPT-2-345M with vanilla AdamW optimizer as a baseline and compare the convergence speed of our SEPARATE-based AdamW optimizer with baseline and PowerSGD [55]. We set different compression ratios of SEPARATE and compare the convergence speed of iteration steps and wall-clock time, respectively in Figure 2 (a) and (b). Figure 2 (a) demonstrates that though high compression ratio results in more steps to reach the same loss, SEPARATE-based optimizer with the proper compression ratio even has a faster convergence speed of iteration steps. Figure 2 (b) shows that SEPARATE can accelerate the training speed up to $2\times$ for GPT-2-345M pre-training by trade-off the compression ratio and one-step cost, compared with baseline and PowerSGD. Thus we think effective designing algorithms based on low-rank gradient and Hessian of model ensures accurate training with lower cost. Moreover, we test the single-step time cost of vanilla AdamW and SEPARATE with different compression ratios on GPT-2-125M and GPT-2-345M. We find that when model scale increases, the effect of SEPARATE is more remarkable, so SEPARATE is significant in training large-scale models.

**Fine-tuning LLAMA2 and Evaluating on Downstream Tasks.** We follow Llama-Accessory [66] and fine-tune LLAMA2-7B[51] model on alpaca-gpt4 [39] dataset for three epochs, and evaluate the ability of fine-tuned model on several downstream tasks including commonsense reasoning, world knowledge, math and code. We report the scores obtained on these evaluation benchmarks in Table 1. We compute the average score (Avg.S, $\uparrow$) and average rank (Avg.R, $\downarrow$) of each method. The results indicate that SEPARATE outperforms other low-rank or low-bit optimizers with error feedback and ZeRO++, even beyond the performance of vanilla Adam optimizer without compression on the comprehensive performance of all the downstream tasks. Moreover, other communication-efficient optimizers have much worse performance than SEPARATE and vanilla Adam, especially on commonsense reasoning tasks. The possible reason for this result is that their last-iteration-based error feedback cannot deal with the accumulated error during the whole process, while SEPARATE is applied with the moving average of the historical error to stabilize the error and error reset mechanism to remove useless information.

## 6.2 ABLATION EXPERIMENTS

We conduct ablation experiments to explore the effect of each component in SEPARATE, including 1)error-feedback technique (Err.Fed.), 2)moving average update of error (Err.Avg.) and 3)error reset mechanism (Err.Re.). For training from scratch, without error-feedback and moving average techniques, the training process becomes unstable and cannot converge. We show the results in Section 6.3. For fine-tuning tasks, we fine-tune the LLAMA2-7B model on alpaca-gpt4 dataset, applying SEPARATE on Adam optimizer with bfloat16 precision. The results are shown in Table 2.

**Error-feedback Technique.** We first study the effect of directly applying the vanilla error-feedback technique on the common random projection compressor of SEPARATE. Comparing SEPARATE1 with SEPARATE2, we can observe that directly applying error-feedback technique even slightly

Table 2: Effects of components in SEPARATE to the performance of LLAMA2-7B fine-tuned on alpaca-gpt4 dataset, including error-feedback technique (Err.Fed.), moving average on error (Err.Avg.) and error reset mechanism (Err.Re.).

| Method | Err.Fed. | Err.Avg. | Err.Re. | GSM8K | MBPP | NQ | Arc-e | Arc-c | PIQA | Avg. |
|---|---|---|---|---|---|---|---|---|---|---|
| SEPARATE1 | False | N/A | N/A | **21.53** | 22.00 | **4.57** | 71.25 | 49.15 | 52.61 | 36.86 |
| SEPARATE2 | True | False | N/A | 20.17 | 21.80 | 4.35 | 69.14 | 51.19 | 52.29 | 36.49 |
| SEPARATE3 | True | True | N/A | 20.92 | **22.00** | 4.49 | 69.49 | **51.53** | 52.23 | 36.77 |
| SEPARATE4 | True | True | 512 | 20.39 | 21.40 | 4.46 | 70.02 | 49.49 | 52.45 | 36.37 |
| SEPARATE5 | True | True | 128 | 20.17 | 21.40 | 4.18 | **72.66** | 49.83 | **52.99** | **36.87** |

Table 3: Effect of different compression ratios of SEPARATE to the performance of LLAMA2-7B fine-tuned on alpaca-gpt4 dataset, where SEPARATE-X means we set the compression ratio at X.

| Method | TriQA | GSM8K | MBPP | NQ | WinoG | Arc-e | Arc-c | PIQA | HellaS | Avg. |
|---|---|---|---|---|---|---|---|---|---|---|
| SEPARATE-8 | 55.43 | 18.88 | 21.00 | 4.13 | 45.78 | **74.07** | **50.85** | 52.83 | **32.43** | 39.49 |
| SEPARATE-16 | 57.18 | **20.17** | 21.40 | 4.18 | 49.25 | 72.66 | 49.83 | **52.99** | 29.73 | **39.71** |
| SEPARATE-64 | **59.00** | 19.03 | 20.40 | 4.54 | 49.8 | 68.43 | 48.47 | 52.88 | 28.59 | 39.02 |
| SEPARATE-128 | 58.93 | 18.65 | **22.40** | **4.79** | **50.28** | 65.78 | 47.46 | 52.67 | 27.79 | 38.75 |

impairs the whole fine-tuning performance, especially on commonsense reasoning tasks such as WinoG, Arc and PIQA. This may be due to the randomness of projection directions. When the random projection directions are far from the dominant directions of Hessian in several continuous iterations, the variance of error will become extremely large and misguide the next iteration. Thus we need another technique to smooth errors and drop the misleading information arising from randomness.

**Moving Average Update of Error.** We use the moving average of the historical error and the current one to replace the direct error from the last iteration, to reduce the instability of vanilla error-feedback technique. Comparing SEPARATE2 and SEPARATE3 we can find that the moving average update of error improves the whole fine-tuning performance, especially on Arc-c and MBPP tasks. However, because of the randomness of the compressor, the accumulated misleading information of inexact directions still impairs the performance compared with SEPARATE1.

**Error Reset Mechanism.** We notice that random projection may generate bad directions far away from the dominant ones of Hessian even with unbiased estimate. Especially compared with quantization methods some bad projections may cause extremely large error. To solve the accumulated bad random information arising from the random projection in the compression error, we design error reset mechanism to reset the accumulated error frequently. Comparing SEPARATE3, SEPARATE4 and SEPARATE5, we find that the appropriate set of error reset frequency $T$ can improve the performance, and too large set of error reset frequency still causes the misleading information accumulation and worse accuracy.

## 6.3 Hyper-parameters Choice of SEPARATE

**Random Seed Choice of Projection.** In the SEPARATE method, we introduce extra randomness due to the random projection. Thus in practice we need to set an extra random seed for commonly generating the same random Gaussian vectors on different devices. In order to clarify that our method works not depending on the choice of seed, we select different common random seeds for projection to repeat training GPT-2-350M with compression ratio 16. The results are shown in Figure 3 (a), where we randomly set seed as 3407, 4396 and 37, and the results make no difference.

$\beta$ **in Moving Average of Error.** SEPARATE method introduces an extra hyper-parameter $\beta$ for moving average update of error. We tested the effect of $\beta$ selection on the convergence speed. We test different $\beta$ in the first 10000 steps of GPT-2-345M training. The results in Figure 3 (b) show that SEPARATE is sensitive to the choice of $\beta$. If we set $\beta = 1$, which means we do not use moving average update of error, the convergence speed shows down remarkably in the first few steps, and in the subsequent iterations it seriously deviates from the optimal dynamic. Setting $\beta = 0.95$ seems to

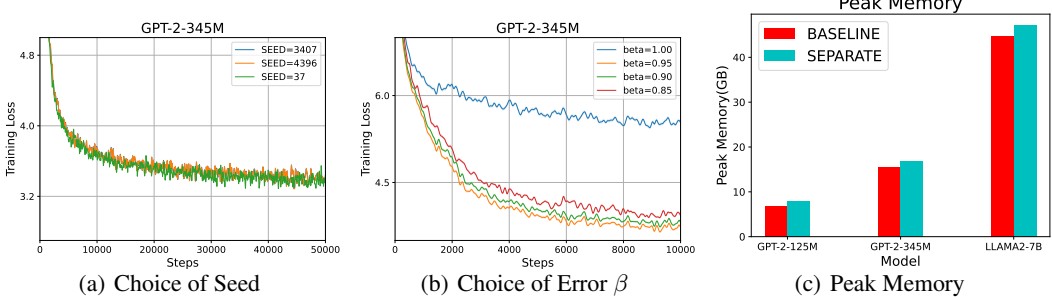

(a) Choice of Seed  (b) Choice of Error $\beta$  (c) Peak Memory

Figure 3: Hyper-parameter choice of SEPARATE and memory cost report. (a) shows the effect of different common random seeds to SEPARATE training with compression ratio 16. (b) shows the effect of different $\beta$ choices of moving average update of error to SEPARATE training with compression ratio 16. (c) shows the peak memory in Adam and SEPARATE training on different models with compression ratio 16.

be acceptable, and when decreasing $\beta$, the convergence speed slows down again. When $\beta \leq 0.80$, the training dynamic becomes unstable with loss value increasing to NaN, which we do not present.

**Compression Ratio.** We conduct experiments to test the effect of the scale of compression ratio on the performance of models. One advantage of SEPARATE is that SEPARATE can set the compression ratio by users themselves instead of 4 or 8 (at most 32) like quantization methods. We set the compression ratio as 8,16,64, and 128, respectively, and tested the performance of the fine-tuned model on downstream tasks. The results are shown in Table 3. The results show that increasing the compression ratio to a certain extent enhances the performance of the model, due to the certain noise improving the generalization of models. Besides, an overlarge compression ratio reduces the performance of models, but the accuracy loss is within the acceptable range because if we compress the gradient 128 times, the communication cost is a quarter of 1-bit quantization methods and the performance is better than it. We also find that with different compression ratios, SEPARATE focuses on different types of tasks, so users can set the ratio adaptively.

## 6.4 MEMORY COST OF SEPARATE

We report the peak memory in Figure 3 (c). Results show the peak memory when training or fine-tuning corresponding models, which demonstrates that the extra memory of SEPARATE is negligible in comparison with the communication cost it saves. Moreover, by expanding the random vector buffer, we can save time in generating random Gaussian vectors, which means we can use some memory cost to exchange computational time in compression and decompression to accelerate the training process.

## 7 CONCLUSION

In this paper, we propose a simple low-rank projection for gradient compression in modern large-scale model training process named SEPARATE. We have carried out theoretical analysis and a lot of experimental verification to illustrate SEPARATE provides an easy way to apply gradient compression to all types of gradient-based optimizers across various training frameworks, while maintaining high-quality training performance and achieving significant compression. SEPARATE holds great potential for future research, particularly in the reduction of the memory cost in the training process. Intuitively, by updating the optimizer state in the low-dimension subspace, it has the potential to save optimizer state memory, and this potential is similar to its ability to save communication overhead. In addition, SEPARATE with adaptive selection of the compression ratio is also worth studying, providing more general framework. In summary, we think SEPARATE is helpful for the further study and popularization of large-scale models.

ACKNOWLEDGMENTS

This work is supported by National Key R&D Program of China (2022ZD0160300) and the NSF China (No.s 62376008 and 62276004).

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

---

**Algorithm 2** SEPARATE: PyTorch-like

---

    random_variable_buffer.**Initialization**() ♯ initialize the variable buffer
    def **Communication_hook** (model.weight): ♯ definition hook function for overlap computation
        projected_grad = **Project** (grad, random_variable_buffer) ♯ random project the grad
        **Communication**(projected_grad)
        grad = **Reproject**(projected_grad, random_variable_buffer )
        random_variable_buffer.**Update**() ♯ update variable buffer with new random variables
    for model.weight in model.weights:
        model.weight.register_hook(**Communication_hook**)
        ♯ register hook for each layer's weight of the model

---

# A   DISCUSSION

In this section, we make a discussion about some related methods for memory-efficient training, the details of our method application, and the robustness. Such a discussion provides us with a clearer understanding of the domain, practicability, and extensibility of our method.

**Compared with Memory-Efficient Training.** Memory-efficient training is another area of efficient training, focusing on how to reduce the memory cost during training process. We notice that random projection is also used for memory reduction, such as Flora for gradient accumulation [19] and GaLore for memory-efficient update in compact space [67]. These approaches seem to show similarity and potential in combination with our method, but we consider different sides of efficient training. It is usually a trade-off because there are key bottlenecks about the extra computational cost for memory reduction. Communication in modern LLMs training framework (e.g., Megatron-LM [48]) overlaps between local computation and global all-reduce. This allows us to tolerate only minimal extra computational preprocessing overhead, lest long synchronization times cause the wall-clock training time to rise. One matrix projection is almost at the limit of tolerance, and SVD is almost intolerable. To the best of our knowledge, only our "simple but efficient operation" can manage to reduce the training wall-clock time.

**Details of SEPARATE Application.** For the common random projection compressor shown in Algorithm 1, we introduce common random vector generators on different nodes of the distributed training clusters. This setup leads our analysis, but it does not incur any additional overhead in actual application, because we can easily set a dedicated common random seed to generate the same random variables on each node during initialization. Moreover, we could easily apply SEPARATE to all kinds of gradient-based optimizers and can be regarded as a plug-in, which can be simply implemented with a few lines of code in off-the-shelf frameworks as shown in Algorithm 2. This means that our method is also the first gradient compression technique that can be seamlessly integrated with FSDP rather than DDP to the best of our knowledge.

**Robustness with Few Parameter Tuning.** We expect to propose a simple and effective gradient compression method that can be efficiently adapted to existing LLMs training frameworks. Therefore, we expect the method to have a certain degree of robustness, that is, it does not depend on the resetting of hyperparameters. In our experiments, all hyperparameters are derived from the default settings of the corresponding model trained under the corresponding framework. In other words, we do not adjust the hyperparameters individually for SEPARATE. We show the hyperparameters setting in Appendix E. In addition, the method introduces the compression ratio. Our variance analysis and the main theorem demonstrate that, provided the selection of the compression ratio aligns with the conditions in our theoretical analysis (Theorem 5.5 and 5.7), our method can exhibit effective performance. This theoretical underpinning ensures that our method is robust within the specified compression ratio. Second, in practical applications, especially for models with a substantial number of parameters, such as those in the millions or billions, we can defaultly set the compression ratio as 16, 32 or 64. As illustrated in Table 3, the performance differences between compression ratios of 16, 32, or 64 times for gradient information are minimal. Consequently, the choice of compression ratio is more influenced by the user's device constraints rather than the intrinsic characteristics of the model.

# B  PROPERTIES OF SEPARATE COMPRESSOR

In this section, we propose several properties of the expectation and variance of the common random projection compressor in SEPARATE. The first analysis of such properties is proposed in the research of distributed optimization theory [64]. Let $\mathbf{a}$ represent the vector for communication and $\tilde{\mathbf{a}}$ represent the estimate one generated by the compressor. In Lemma 4.1 and Lemma 4.2, we show that $\tilde{\mathbf{a}}$ is an unbiased estimator, and the variance of $\tilde{\mathbf{a}}$ can be $\mathrm{tr}(\mathbf{A})$-bounded under arbitrary matrix $\mathbf{A}$-norms, respectively.

**Lemma B.1.** *$\tilde{\mathbf{a}}$ is an unbiased estimator of* $\mathbf{a}$,

$$\mathbb{E}_{\boldsymbol{\xi}_1,\cdots\boldsymbol{\xi}_m}\tilde{\mathbf{a}} = \mathbf{a}. \tag{14}$$

*Proof of Lemma 4.1.*

$$
\begin{aligned}
\mathbb{E}_{\boldsymbol{\xi}_1,\cdot,\boldsymbol{\xi}_m}\tilde{\mathbf{a}} &= \mathbb{E}_{\boldsymbol{\xi}_1,\cdots,\boldsymbol{\xi}_m}\left[\frac{1}{m}\sum_{i=1}^m \langle \mathbf{a},\boldsymbol{\xi}_i\rangle \cdot \boldsymbol{\xi}_i\right] \\
&= \mathbb{E}_{\boldsymbol{\xi}_1}\boldsymbol{\xi}_1\boldsymbol{\xi}_1^\top \mathbf{a} = \mathbf{I}\mathbf{a} \\
&= \mathbf{a}
\end{aligned}
\tag{15}
$$

$\square$

**Lemma B.2.** *The variance of $\tilde{\mathbf{a}}$ under norm $\|\cdot\|_{\mathbf{A}}$, where $\mathbf{A}$ is a given positive semi-definite symmetric matrix, can be bounded by* $\frac{3\mathrm{tr}(\mathbf{A})}{m}\|\mathbf{a}\|^2 - \frac{1}{m}\|\mathbf{a}\|_{\mathbf{A}}^2$,

$$\mathbb{E}_{\boldsymbol{\xi}_1,\cdots,\boldsymbol{\xi}_m}\|\tilde{\mathbf{a}}-\mathbf{a}\|_{\mathbf{A}}^2 \le \frac{3\mathrm{tr}(\mathbf{A})}{m}\|\mathbf{a}\|^2 - \frac{1}{m}\|\mathbf{a}\|_{\mathbf{A}}^2. \tag{16}$$

*Proof of Lemma 4.2.* For the simplicity of notation, we use $\mathbb{E}_{\boldsymbol{\xi}}$ to denote $\mathbb{E}_{\boldsymbol{\xi}_1,\cdots,\boldsymbol{\xi}_m}$.

$$
\begin{aligned}
\mathbb{E}_{\boldsymbol{\xi}}\|\tilde{\mathbf{a}}-\mathbf{a}\|_{\mathbf{A}}^2 &= \mathbb{E}_{\boldsymbol{\xi}}\left\|\frac{1}{m}\sum_{i=1}^m (\langle\mathbf{a},\boldsymbol{\xi}\rangle\cdot\boldsymbol{\xi}-\mathbf{a})\right\|_{\mathbf{A}}^2 \\
&= \mathbb{E}_{\boldsymbol{\xi}}\left[\frac{1}{m^2}\sum_{i=1}^m \left(\mathbf{a}^\top\boldsymbol{\xi}_i\boldsymbol{\xi}_i^\top\mathbf{A}\boldsymbol{\xi}_i\boldsymbol{\xi}_i^\top\mathbf{a}-\mathbf{a}^\top\mathbf{A}\mathbf{a}\right)\right] \\
&= \frac{1}{m}\mathbb{E}_{\boldsymbol{\xi}_1}\mathbf{a}^\top\boldsymbol{\xi}_1\boldsymbol{\xi}_1^\top\mathbf{A}\boldsymbol{\xi}_1\boldsymbol{\xi}_1^\top\mathbf{a}-\frac{1}{m}\|\mathbf{a}\|_{\mathbf{A}}^2.
\end{aligned}
\tag{17}
$$

Let $\mathbf{A} = \mathbf{U}^\top\mathbf{D}\mathbf{U}$ be the eigenvalue decomposition of $\mathbf{A}$ where $\mathbf{D} = \mathrm{diag}\{b_1,\cdots,b_d\}$ is a diagonal matrix, and $\boldsymbol{\zeta} = \mathbf{U}\boldsymbol{\xi}_1$ be a linear transformation of the random variable $\boldsymbol{\xi}_1$. We have

$$
\begin{aligned}
\mathbb{E}_{\boldsymbol{\xi}_1}\left[\boldsymbol{\xi}_1\boldsymbol{\xi}_1^\top\mathbf{A}\boldsymbol{\xi}_1\boldsymbol{\xi}_1^\top\right] &\overset{a}{=} \mathbb{E}_{\boldsymbol{\zeta}}\left[\mathbf{U}^\top\boldsymbol{\zeta}\boldsymbol{\zeta}^\top\mathbf{D}\boldsymbol{\zeta}\boldsymbol{\zeta}^\top\mathbf{U}\right] \\
&= \mathbf{U}^\top\mathbb{E}_{\boldsymbol{\zeta}}\left[\sum_{i=1}^d b_i\zeta_i^2\cdot\boldsymbol{\zeta}\boldsymbol{\zeta}^\top\right]\mathbf{U} \\
&\overset{b}{=} \mathbf{U}^\top\left(\sum_{i=1}^d b_i\cdot\mathbf{I}+2\mathbf{D}\right)\mathbf{U} \\
&\overset{c}{=} \mathrm{tr}(\mathbf{A})\cdot\mathbf{I}+2\mathbf{A} \\
&\preceq 3\mathrm{tr}(\mathbf{A})\cdot\mathbf{I}.
\end{aligned}
\tag{18}
$$

In $\overset{a}{=}$, we use $\boldsymbol{\zeta} \sim N(0,\mathbf{I}_d)$ based on the rotational invariance of the standard Gaussian distribution. In $\overset{b}{=}$, we use the second and forth moment of standard Gaussian variables: $\mathbb{E}\zeta_i^2 = 1$ and $\mathbb{E}\zeta_i^4 = 3$. In $\overset{c}{=}$, we use $\mathrm{tr}(\mathbf{U}^\top\mathbf{D}\mathbf{U}) = \mathrm{tr}(\mathbf{U}^\top\mathbf{U}\mathbf{D}) = \mathrm{tr}(\mathbf{D})$. The last inequality of (18) is due to $\mathrm{tr}(\mathbf{A})\cdot\mathbf{I}\succeq\mathbf{A}$. Combining (18) and (17), we have

$$\mathbb{E}_{\boldsymbol{\xi}_1,\cdots,\boldsymbol{\xi}_m}\|\tilde{\mathbf{a}}-\mathbf{a}\|_{\mathbf{A}}^2 \le \frac{3\mathrm{tr}(\mathbf{A})}{m}\|\mathbf{a}\|^2 - \frac{1}{m}\|\mathbf{a}\|_{\mathbf{A}}^2. \tag{19}$$

$\square$

## C   DEFERRED PROOF IN SECTION 5

### C.1   USEFUL LEMMAS

In this Section, we propose several useful lemmas for the proof of our main theorem in Section 5.

**Lemma C.1.** *For the gradient $\tilde{\mathbf{h}}_n^k = \frac{1}{m} \sum_{i=1}^m \tilde{p}_i \cdot \boldsymbol{\xi}_i$ where $\tilde{p}_i = \frac{1}{N} \sum_{n=1}^N p_{i,n}$ is the global average of $p_{i,n}$ in each machine after communication in Algorithm 1, we have*

$$\tilde{\mathbf{h}}^k = \mathbf{g}^k + \frac{1}{\beta} \left( \mathbf{e}^k - \mathbf{e}^{k+1} \right),\tag{20}$$

*where $\tilde{\mathbf{h}}^k = \frac{1}{N} \sum_{n=1}^N \tilde{\mathbf{h}}_n^k$, $\mathbf{g}^k = \frac{1}{N} \sum_{n=1}^N \mathbf{g}_n^k$ and $\mathbf{e}^k = \frac{1}{N} \sum_{n=1}^N \mathbf{e}_n^k$.*

*Proof of Lemma C.1.* For the estimate $\tilde{\mathbf{h}}^k$ defined in Lemma C.1, we have

$$\begin{aligned}
\tilde{\mathbf{h}}^k &= \frac{1}{N} \sum_{n=1}^N \tilde{\mathbf{h}}_n^k \\
&= \frac{1}{N} \sum_{n=1}^N \frac{1}{m} \sum_{i=1}^m \frac{1}{N} \sum_{n=1}^N \langle \mathbf{h}_n^k, \boldsymbol{\xi}_i \rangle \cdot \boldsymbol{\xi}_i \\
&= \frac{1}{m} \sum_{i=1}^m \frac{1}{N} \sum_{n=1}^N \langle \mathbf{h}_n^k, \boldsymbol{\xi}_i \rangle \cdot \boldsymbol{\xi}_i \\
&= \frac{1}{N} \sum_{n=1}^N \left( \mathbf{h}_n^k + \frac{1}{m} \sum_{i=1}^m \langle \mathbf{h}_n^k, \boldsymbol{\xi}_i \rangle \cdot \boldsymbol{\xi}_i - \mathbf{h}_n^k \right) \\
&= \frac{1}{N} \sum_{n=1}^N \mathbf{g}_n^k + \mathbf{e}_n^k - \delta_n^k
\end{aligned}\tag{21}$$

where $\delta_n^k = \mathbf{h}_n^k - \frac{1}{m} \sum_{i=1}^m \langle \mathbf{h}_n^k, \boldsymbol{\xi}_i \rangle \cdot \boldsymbol{\xi}_i$. Taking average of $\delta_n^k$, we have

$$\begin{aligned}
\delta^k &= \frac{1}{N} \sum_{n=1}^N \left( \mathbf{h}_n^k - \frac{1}{m} \sum_{i=1}^m \langle \mathbf{h}_n^k, \boldsymbol{\xi}_i \rangle \cdot \boldsymbol{\xi}_i \right) \\
&= \mathbf{h}^k - \tilde{\mathbf{h}}^k \\
&= \frac{1}{\beta} \left( \mathbf{e}^{k+1} - (1 - \beta)\mathbf{e}^k \right),
\end{aligned}\tag{22}$$

where the last equality uses $\mathbf{e}^{k+1} = (1 - \beta)\mathbf{e}^k + \beta(\mathbf{h}^k - \tilde{\mathbf{h}}^k)$ which is the average of the iteration of $\mathbf{e}_n^{k+1}$ in Algorithm 1. Combining the two equalities above, we have

$$\tilde{\mathbf{h}}^k = \mathbf{g}^k + \mathbf{e}^k - \delta^k = \mathbf{g}^k + \frac{1}{\beta} \left( \mathbf{e}^k - \mathbf{e}^{k+1} \right).\tag{23}$$

We complete the proof of Lemma C.1. $\qquad\square$

**Lemma C.2.** *Suppose the setting in Theorem 5.5 hold, and we use $\tilde{\mathbf{h}}^k$ in Algorithm 1 to update the model weight $\boldsymbol{\theta}^k$ in SGD as*

$$\boldsymbol{\theta}^{k+1} = \boldsymbol{\theta}^k - \eta \tilde{\mathbf{h}}^k.\tag{24}$$

*Then we have*

$$\mathbb{E} \left\| \sum_{i=0}^k (\tilde{\mathbf{h}}^i - \mathbf{g}^i) \right\| \le \frac{1 + (1 - \beta)^{k+1}}{\beta} c_1.\tag{25}$$

*Proof of Lemma C.2.* Consider the expectation of $\|\mathbf{e}^{k+1}\|$ about the random Gaussian variables $\boldsymbol{\xi}_1, \cdots, \boldsymbol{\xi}_m$ and the stochastic mini-batch as below

$$\mathbb{E}\|\mathbf{e}^{k+1}\| = \mathbb{E} \left[ \mathbb{E}_{\boldsymbol{\xi}_1, \cdots, \boldsymbol{\xi}_m} \|\mathbf{e}^{k+1}\| \right].\tag{26}$$

First we have

$$
\begin{aligned}
\mathbb{E}_{\boldsymbol{\xi}_1,\cdots,\boldsymbol{\xi}_m} \|\mathbf{e}^{k+1}\| &= \mathbb{E}_{\boldsymbol{\xi}_1,\cdots,\boldsymbol{\xi}_m} \|(1-\beta)\mathbf{e}^k + \beta(\mathbf{h}^k - \tilde{\mathbf{h}}^k)\| \\
&\leq (1-\beta)\mathbb{E}_{\boldsymbol{\xi}_1,\cdots,\boldsymbol{\xi}_m}\|\mathbf{e}^k\| + \mathbb{E}_{\boldsymbol{\xi}_1,\cdots,\boldsymbol{\xi}_m}\|\beta(\mathbf{h}^k - \tilde{\mathbf{h}}^k)\| \\
&= (1-\beta)\|\mathbf{e}^k\|,
\end{aligned}
\tag{27}
$$

where the first inequality uses $\tilde{\mathbf{h}}^k$ is the unbiased estimate of $\mathbf{h}^k$. Thus we have

$$
\mathbb{E}\|\mathbf{e}^{k+1}\| \leq (1-\beta)\mathbb{E}\|\mathbf{e}^k\| \leq (1-\beta)^{k+1}\|\mathbf{e}^0\| \leq (1-\beta)^{k+1}c_1.
\tag{28}
$$

Considering the accumulated bias of $\tilde{\mathbf{h}}^k$ and $\mathbf{h}^k$, based on Lemma C.1 we have

$$
\mathbb{E}\left\|\sum_{i=0}^{k}(\tilde{\mathbf{h}}^i - \mathbf{g}^i)\right\| = \frac{1}{\beta}\mathbb{E}\left\|\sum_{i=0}^{k}\mathbf{e}^i - \mathbf{e}^{i+1}\right\| \leq \frac{1}{\beta}\left(\mathbb{E}\|\mathbf{e}^{k+1}\| + c_1\right) \leq \frac{1+(1-\beta)^{k+1}}{\beta}c_1.
\tag{29}
$$

$\square$

**Lemma C.3.** *Suppose the setting in Theorem 5.7 hold, and we use $\tilde{\mathbf{h}}^k$ in Algorithm 1 to update te model weight $\boldsymbol{\theta}^k$ in Adam-Type optimizers as (8). Then consider the following two sequences $\{\mathbf{m}^k\}_{k=0}^K$ and $\{\tilde{\mathbf{m}}^k\}_{k=0}^K$ as:*

$$
\begin{aligned}
\mathbf{m}^k &= (1-\beta_1)\mathbf{m}^{k-1} + \beta_1\mathbf{g}^k, \\
\tilde{\mathbf{m}}^k &= (1-\beta_1)\tilde{\mathbf{m}}^{k-1} + \beta_1\tilde{\mathbf{h}}^k.
\end{aligned}
\tag{30}
$$

*Assume that for the sequence $\{\boldsymbol{\eta}^k\}_{k=0}^K$, each element of $\boldsymbol{\eta}^k$ satisfies*

$$
\eta c_l \leq \eta_i^k \leq \eta c_u, \qquad |\eta_i^k - \eta_i^{k-1}| \leq \eta\beta_1(1-\beta_1)^{K-k}c_u, \forall i \in [d], k \in [K].
\tag{31}
$$

*Then we have*

$$
\mathbb{E}\left\|\sum_{i=0}^{K}\boldsymbol{\eta}^i \circ \left(\tilde{\mathbf{m}}^i - \mathbf{m}^i\right)\right\| \leq \frac{2c_1 c_u \eta\sqrt{d}}{\beta}.
\tag{32}
$$

*Proof of Lemma C.3.* First we extend the iterations of $\{\mathbf{m}^k\}_{k=0}^K$ and $\{\tilde{\mathbf{m}}^k\}_{k=0}^K$:

$$
\mathbf{m}^k = (1-\beta_1)\mathbf{m}^{k-1} + \beta_1\mathbf{g}^k = (1-\beta_1)^{k-1}\mathbf{m}^0 + \beta_1\sum_{t=1}^{k}(1-\beta_1)^{k-t}\mathbf{g}^t,
\tag{33}
$$

and

$$
\tilde{\mathbf{m}}^k = (1-\beta_1)\tilde{\mathbf{m}}^{k-1} + \beta_1\tilde{\mathbf{h}}^k = (1-\beta_1)^{k-1}\mathbf{m}^0 + \beta_1\sum_{t=1}^{k}(1-\beta_1)^{k-t}\tilde{\mathbf{h}}^t.
\tag{34}
$$

Based on Lemma C.1, we have

$$
\sum_{i=0}^{k}\boldsymbol{\eta}^i \circ \left(\tilde{\mathbf{m}}^i - \mathbf{m}^i\right) \leq \frac{\beta_1}{\beta}\sum_{i=0}^{k}\boldsymbol{\eta}^i \circ \left(\sum_{t=0}^{i}(1-\beta_1)^{i-t}(\mathbf{e}^t - \mathbf{e}^{t+1})\right).
\tag{35}
$$

Then we element-wisely analyse the upper bound of the accumulated error as below. We use the non-blackbody letters to represent the element of vectors for convenience with some symbolic abuse.

$$
\begin{aligned}
&\left|\frac{\beta_1}{\beta}\sum_{i=0}^{k}\eta^i\sum_{t=0}^{i}(1-\beta_1)^{i-t}(e^t - e^{t+1})\right| \\
&= \left|\frac{\beta_1}{\beta}\sum_{t=0}^{k}\left(\sum_{i=t}^{k}(\eta^i - \eta^{i-1})(1-\beta_1)^{i-t} - \eta^{k-1}(1-\beta^1)^{k-t}\right)e^t\right|.
\end{aligned}
\tag{36}
$$

Then using Lemma C.2, we take the expectation of the accumulated error as below:

$$
\begin{aligned}
\mathbb{E} &\left\| \sum_{i=0}^{k} \boldsymbol{\eta}^i \circ \left( \tilde{\mathbf{m}}^i - \mathbf{m}^i \right) \right\| \\
&\leq \sqrt{d} \cdot \mathbb{E} \left\| \sum_{i=0}^{k} \boldsymbol{\eta}^i \circ \left( \tilde{\mathbf{m}}^i - \mathbf{m}^i \right) \right\|_\infty \\
&\leq \sqrt{d} \cdot \mathbb{E} \sup_{n \in [d]} \left| \frac{\beta_1}{\beta} \sum_{t=0}^{k} \left( \sum_{i=t}^{k-1} (\eta_n^i - \eta_n^{i-1})(1-\beta_1)^{i-t} - \eta_n^{k-1}(1-\beta^1)^{k-t} \right) e_n^t \right| \\
&\leq \frac{\beta_1 c_1 c_u \eta \sqrt{d}}{\beta} \mathbb{E} \left| \beta_1 \sum_{t=1}^{k-1} t(1-\beta_1)^t + \sum_{t=1}^{k-1} (1-\beta_1)^t \right| \\
&\leq \frac{2 c_1 c_u \eta \sqrt{d}}{\beta}.
\end{aligned}
\tag{37}
$$

For the first inequality we use the property of vector norm that $\| \cdot \| \leq \sqrt{d} \| \cdot \|_\infty$. For the second inequality we use the definition of vector's infinite norm. For the third inequality we use the assumption that $|\eta^i - \eta^{i-1}| \leq \eta \beta_1 (1-\beta_1)^{K-i} c_u, \forall i \in [K]$ to obtain the upper bound of each element, and use Lemma C.2 to estimate the upper bound of $\mathbb{E}|e_n^t|$ that $\mathbb{E}|e_n^t| \leq \mathbb{E}|e_n^0| \leq c_1$. For the forth inequality we use the summation of series to obtain that $\beta_1 \sum_{t=1}^{k-1} t(1-\beta_1)^t + \sum_{t=1}^{k-1}(1-\beta_1)^t \leq \frac{2}{\beta_1}$. Thus we have

$$
\mathbb{E} \left\| \sum_{i=0}^{k} \boldsymbol{\eta}^i \circ \left( \tilde{\mathbf{m}}^i - \mathbf{m}^i \right) \right\| \leq \frac{2 c_1 c_u \eta \sqrt{d}}{\beta}.
\tag{38}
$$

□

**Lemma C.4.** *Consider the moving average iteration like AdamType as*

$$
\mathbf{m}^k = (1-\beta)\mathbf{m}^{k-1} + \beta \mathbf{g}^k,
\tag{39}
$$

*where $\mathbf{g}^k = \nabla f(\boldsymbol{\theta}^k) + \boldsymbol{\xi}^k$ is the stochastic gradient with $\mathbb{E}\boldsymbol{\xi}^k = 0$ and $\mathbb{E}\|\boldsymbol{\xi}^k\|^2 \leq \sigma^2$. Then we have*

$$
\mathbb{E} \left\| \mathbf{m}^k - \nabla f(\boldsymbol{\theta}^k) \right\|^2 \leq (1-\beta)\mathbb{E} \left\| \mathbf{m}^{k-1} - \nabla f(\boldsymbol{\theta}^{k-1}) \right\|^2 + \frac{(1-\beta)^2 L^2}{\beta} \mathbb{E} \left\| \boldsymbol{\theta}^{k-1} - \boldsymbol{\theta}^k \right\|^2 + \beta^2 \sigma^2.
\tag{40}
$$

*Proof of Lemma C.4.* Based on the iteration of $\mathbf{m}$ we have

$$
\mathbf{m}^k - \nabla f(\boldsymbol{\theta}^k) = (1-\beta)\left(\mathbf{m}^{k-1} - \nabla f(\boldsymbol{\theta}^{k-1})\right) + (1-\beta)\left(\nabla f(\boldsymbol{\theta}^{k-1}) - \nabla f(\boldsymbol{\theta}^k)\right) + \beta\left(\mathbf{g}^k - \nabla f(\boldsymbol{\theta}^k)\right).
\tag{41}
$$

Then taking expectation of both sides we have

$$
\begin{aligned}
\mathbb{E} \left\| \mathbf{m}^k - \nabla f(\boldsymbol{\theta}^k) \right\|^2 &= (1-\beta)^2 \mathbb{E} \left\| \mathbf{m}^{k-1} - \nabla f(\boldsymbol{\theta}^{k-1}) \right\|^2 + (1-\beta)^2 \mathbb{E} \left\| \nabla f(\boldsymbol{\theta}^{k-1}) - \nabla f(\boldsymbol{\theta}^{k-1}) \right\|^2 \\
&\quad + \beta^2 \sigma^2 + 2(1-\beta)^2 \mathbb{E} \left( \langle \mathbf{m}^{k-1} - \nabla f(\boldsymbol{\theta}^{k-1}), \nabla f(\boldsymbol{\theta}^{k-1}) - \nabla f(\boldsymbol{\theta}^{k-1}) \rangle \right) \\
&\leq (1-\beta)^2 (1+a) \mathbb{E} \left\| \mathbf{m}^{k-1} - \nabla f(\boldsymbol{\theta}^{k-1}) \right\|^2 \\
&\quad + (1-\beta)^2 \left( 1 + \frac{1}{a} \right) \mathbb{E} \left\| \nabla f(\boldsymbol{\theta}^{k-1}) - \nabla f(\boldsymbol{\theta}^{k-1}) \right\|^2 + \beta^2 \sigma^2 \\
&\leq (1-\beta) \mathbb{E} \left\| \mathbf{m}^{k-1} - \nabla f(\boldsymbol{\theta}^{k-1}) \right\|^2 \\
&\quad + \frac{(1-\beta)^2}{\beta} \mathbb{E} \left\| \nabla f(\boldsymbol{\theta}^{k-1}) - \nabla f(\boldsymbol{\theta}^{k-1}) \right\|^2 + \beta^2 \sigma^2 \\
&\leq (1-\beta) \mathbb{E} \left\| \mathbf{m}^{k-1} - \nabla f(\boldsymbol{\theta}^{k-1}) \right\|^2 + \frac{(1-\beta)^2 L^2}{\beta} \mathbb{E} \left\| \boldsymbol{\theta}^{k-1} - \boldsymbol{\theta}^k \right\|^2 + \beta^2 \sigma^2.
\end{aligned}
\tag{42}
$$

□

## C.2 PROOF OF THEOREM 5.5

*Proof.* Consider the two SGD-based iteration $\{\boldsymbol{\theta}^k\}_{k=1}^K$ as:

$$\boldsymbol{\theta}^{k+1} = \boldsymbol{\theta}^k - \eta\mathbf{g}^k = \boldsymbol{\theta}^0 - \eta\sum_{i=0}^k \mathbf{g}^i \tag{43}$$

and $\{\tilde{\boldsymbol{\theta}}^k\}_{k=1}^K$ as:

$$\tilde{\boldsymbol{\theta}}^{k+1} = \tilde{\boldsymbol{\theta}}^k - \eta\tilde{\mathbf{h}}^k = \boldsymbol{\theta}^0 - \eta\sum_{i=0}^k \mathbf{g}^i + \eta\sum_{i=0}^k \left(\mathbf{g}^i - \tilde{\mathbf{h}}^i\right). \tag{44}$$

For $k \in [K]$, due to the $L$-smooth assumption of the objective $f$ and Lemma C.1, we have

$$\mathbb{E}\left\|\nabla f(\tilde{\boldsymbol{\theta}}^k) - \nabla f(\boldsymbol{\theta}^k)\right\| \le L\mathbb{E}\left\|\tilde{\boldsymbol{\theta}}^k - \boldsymbol{\theta}^k\right\| = \eta L\mathbb{E}\left\|\sum_{i=0}^{k-1}\left(\mathbf{g}^i - \tilde{\mathbf{h}}^i\right)\right\| \le \frac{(1+(1-\beta)^k)\eta Lc_1}{\beta}. \tag{45}$$

Next, we write the second-order Taylor expansion of $f(\boldsymbol{\theta}^{k+1})$ at $\boldsymbol{\theta}^k$ as below:

$$f(\boldsymbol{\theta}^{k+1}) \le f(\boldsymbol{\theta}^k) + \langle\nabla f(\boldsymbol{\theta}^k), \boldsymbol{\theta}^{k+1} - \boldsymbol{\theta}^k\rangle + \frac{1}{2}\langle\mathbf{A}(\boldsymbol{\theta}^{k+1} - \boldsymbol{\theta}^k), \boldsymbol{\theta}^{k+1} - \boldsymbol{\theta}^k\rangle. \tag{46}$$

Taking expectation of (46) with condition on the iteration before $k$ and random Gaussian variables to both sides of (46), using Lemma 4.1, Lemma 4.2 and Assumption 5.4, we have

$$
\begin{aligned}
\mathbb{E}f(\boldsymbol{\theta}^{k+1}) &\le \mathbb{E}\left[f(\boldsymbol{\theta}^k) + \langle\nabla f(\boldsymbol{\theta}^k), \boldsymbol{\theta}^{k+1} - \boldsymbol{\theta}^k\rangle + \eta^2\left(\frac{3\mathrm{tr}(\nabla^2 f(\boldsymbol{\theta}^k))}{2m}\left\|\nabla f(\tilde{\boldsymbol{\theta}}^k)\right\|^2 + \left\|\nabla f(\tilde{\boldsymbol{\theta}}^k)\right\|_{\mathbf{A}}^2\right)\right]\\
&\le \mathbb{E}f(\boldsymbol{\theta}^k) - \eta\mathbb{E}\left\langle\nabla f(\boldsymbol{\theta}^k), \nabla f(\tilde{\boldsymbol{\theta}}^k)\right\rangle + \eta\mathbb{E}\left\langle\nabla f(\boldsymbol{\theta}^k), \nabla f(\tilde{\boldsymbol{\theta}}^k) - \nabla f(\boldsymbol{\theta}^k)\right\rangle\\
&\quad + \eta^2\left(\frac{3\mathrm{tr}(\mathbf{A})}{2m} + L\right)\mathbb{E}\left\|\nabla f(\tilde{\boldsymbol{\theta}}^k)\right\|^2\\
&\overset{a}{\le} \mathbb{E}f(\boldsymbol{\theta}^k) - \eta\left\|\nabla f(\tilde{\boldsymbol{\theta}}^k)\right\|^2 + \eta\mathbb{E}\left\langle\nabla f(\boldsymbol{\theta}^k), \nabla f(\tilde{\boldsymbol{\theta}}^k) - \nabla f(\boldsymbol{\theta}^k)\right\rangle\\
&\quad + \eta\mathbb{E}\left\langle\nabla f(\tilde{\boldsymbol{\theta}}^k) - \nabla f(\boldsymbol{\theta}^k), \nabla f(\tilde{\boldsymbol{\theta}}^k)\right\rangle + \eta^2\cdot\frac{5\mathrm{tr}(\mathbf{A})}{2m}\mathbb{E}\left\|\nabla f(\tilde{\boldsymbol{\theta}}^k)\right\|^2\\
&\le \mathbb{E}f(\boldsymbol{\theta}^k) - \eta\left\|\nabla f(\tilde{\boldsymbol{\theta}}^k)\right\|^2 + \mathcal{O}\left(\eta\left(\frac{(1+(1-\beta)^k)\eta Lc_1}{\beta}\sqrt{d}c_\infty\right)\right)\\
&\quad + \eta^2\cdot\frac{5\mathrm{tr}(\mathbf{A})}{2m}\left(\left\|\nabla f(\tilde{\boldsymbol{\theta}}^k)\right\|^2 + \frac{\sigma^2}{N}\right)
\end{aligned}
\tag{47}
$$

where in $\overset{a}{\le}$ we use $m \le \frac{\mathrm{tr}(\mathbf{A})}{L}$. Then, using $\eta \le \frac{m}{4\mathrm{tr}(\mathbf{A})}$, we have

$$\frac{1}{K}\sum_{k=0}^K \mathbb{E}\left\|\nabla f(\tilde{\boldsymbol{\theta}}^k)\right\|^2 \le \frac{8\left(f(\tilde{\boldsymbol{\theta}}^0) - f(\tilde{\boldsymbol{\theta}}^*)\right)}{3\eta K} + \mathcal{O}\left(\frac{(2-\beta)\eta Lc_1}{\beta}\sqrt{d}c_\infty\right) + \mathcal{O}\left(\frac{\eta\sigma^2}{N}\right). \tag{48}$$

By letting $\eta = \mathcal{O}\left(d^{-1/2}\epsilon^2\right)$, $K = \Omega\left(d^{1/2}\epsilon^{-4}\right)$. we have

$$\frac{1}{K}\sum_{k=0}^K \mathbb{E}\left\|\nabla f(\tilde{\boldsymbol{\theta}}^k)\right\|^2 \le \mathcal{O}\left(\epsilon^2\left(f(\tilde{\boldsymbol{\theta}}^0) - f(\tilde{\boldsymbol{\theta}}^*) + \frac{(2-\beta)Lc_1}{\beta}c_\infty + \frac{\sigma^2}{N}\right)\right) = \mathcal{O}\left(\epsilon^2\right). \tag{49}$$

Thus we finish the proof of Theorem 5.5. $\qquad\square$

## C.3 PROOF OF THEOREM 5.7

*Proof.* We also consider two AdamType-based iteration $\{\boldsymbol{\theta}^k\}_{k=1}^K$ as:

$$\boldsymbol{\theta}^{k+1} = \boldsymbol{\theta}^k - \boldsymbol{\eta}^k \circ \mathbf{m}^k = \boldsymbol{\theta}^0 - \sum_{i=0}^k \boldsymbol{\eta}^i \circ \mathbf{m}^i, \quad (50)$$

where $\mathbf{m}^k = (1 - \beta_1)\mathbf{m}^{k-1} + \beta_1 \mathbf{g}^k$, and $\{\tilde{\boldsymbol{\theta}}^k\}_{k=1}^K$ as:

$$\tilde{\boldsymbol{\theta}}^{k+1} = \tilde{\boldsymbol{\theta}}^k - \boldsymbol{\eta}^k \circ \tilde{\mathbf{m}}^k = \boldsymbol{\theta}^0 - \sum_{i=0}^k \boldsymbol{\eta}^i \circ \tilde{\mathbf{m}}^i = \boldsymbol{\theta}^0 - \sum_{i=0}^k \boldsymbol{\eta}^i \circ \mathbf{m}^i + \sum_{i=0}^k \boldsymbol{\eta}^i \circ (\mathbf{m}^i - \tilde{\mathbf{m}}^i), \quad (51)$$

where $\tilde{\mathbf{m}}^k = (1 - \beta_1)\tilde{\mathbf{m}}^{k-1} + \beta_1 \mathbf{h}^k$. Based on Lemma C.3, we have

$$\mathbb{E}\left\|\nabla f(\tilde{\boldsymbol{\theta}}^k) - \nabla f(\boldsymbol{\theta}^k)\right\| \le L\mathbb{E}\left\|\tilde{\boldsymbol{\theta}}^k - \boldsymbol{\theta}^k\right\| = L\mathbb{E}\left\|\sum_{i=0}^k \boldsymbol{\eta}^i \circ (\mathbf{m}^i - \tilde{\mathbf{m}}^i)\right\| \le \frac{2Lc_1 c_u \eta \sqrt{d}}{\beta}, \quad (52)$$

and

$$\mathbb{E}\left\|\tilde{\boldsymbol{\theta}}^k - \boldsymbol{\theta}^k\right\|^2 = \mathbb{E}\left\|\sum_{i=0}^k \boldsymbol{\eta}^i \circ (\mathbf{m}^i - \tilde{\mathbf{m}}^i)\right\|^2 \le \frac{4c_1^2 c_u^2 \eta^2 d}{\beta^2}. \quad (53)$$

Based on the $L$-smooth assumption of the objective $f$, we have

$$
\begin{aligned}
\mathbb{E}f(\boldsymbol{\theta}^{k+1}) &\le \mathbb{E}f(\boldsymbol{\theta}^k) + \mathbb{E}\left\langle \nabla f(\boldsymbol{\theta}^k), \boldsymbol{\theta}^{k+1} - \boldsymbol{\theta}^k \right\rangle + \frac{L}{2}\mathbb{E}\left\|\boldsymbol{\theta}^{k+1} - \boldsymbol{\theta}^k\right\|^2 \\
&= \mathbb{E}f(\boldsymbol{\theta}^k) - \mathbb{E}\left\langle \nabla f(\boldsymbol{\theta}^k), \boldsymbol{\eta}^k \circ \mathbf{m}^k \right\rangle + \frac{L}{2}\mathbb{E}\left\|\boldsymbol{\theta}^{k+1} - \boldsymbol{\theta}^k\right\|^2 \\
&= \mathbb{E}f(\boldsymbol{\theta}^k) - \mathbb{E}\left\langle \nabla f(\tilde{\boldsymbol{\theta}}^k), \boldsymbol{\eta}^k \circ \mathbf{m}^k \right\rangle + \frac{L}{2}\mathbb{E}\left\|\boldsymbol{\theta}^{k+1} - \boldsymbol{\theta}^k\right\|^2 \\
&\quad + \mathbb{E}\left\langle \nabla f(\tilde{\boldsymbol{\theta}}^k) - \nabla f(\boldsymbol{\theta}^k), \boldsymbol{\eta}^k \circ \mathbf{m}^k \right\rangle \\
&\le \mathbb{E}f(\boldsymbol{\theta}^k) - \mathbb{E}\left\langle \nabla f(\tilde{\boldsymbol{\theta}}^k), \boldsymbol{\eta}^k \circ \mathbf{m}^k \right\rangle + \frac{L}{2}\mathbb{E}\left\|\boldsymbol{\theta}^{k+1} - \boldsymbol{\theta}^k\right\|^2 + \frac{2Lc_1 c_u c_\infty \eta^2 d}{\beta} \\
&= \mathbb{E}f(\boldsymbol{\theta}^k) + \frac{1}{2}\mathbb{E}\left\|\sqrt{\boldsymbol{\eta}^k} \circ \left(\nabla f(\tilde{\boldsymbol{\theta}}^k) - \mathbf{m}^k\right)\right\|^2 - \frac{1}{2}\mathbb{E}\left\|\sqrt{\boldsymbol{\eta}^k} \circ \nabla f(\tilde{\boldsymbol{\theta}}^k)\right\|^2 \\
&\quad - \frac{1}{2}\mathbb{E}\left\|\sqrt{\boldsymbol{\eta}^k} \circ \mathbf{m}^k\right\|^2 + \frac{L}{2}\mathbb{E}\left\|\boldsymbol{\eta}^k \circ \mathbf{m}^k\right\|^2 + \frac{2Lc_1 c_u c_\infty \eta^2 d}{\beta} \\
&\le \mathbb{E}f(\boldsymbol{\theta}^k) + \frac{c_u \eta}{2}\mathbb{E}\left\|\nabla f(\tilde{\boldsymbol{\theta}}^k) - \mathbf{m}^k\right\|^2 - \frac{c_l \eta}{2}\mathbb{E}\left\|\nabla f(\tilde{\boldsymbol{\theta}}^k)\right\|^2 \\
&\quad + \left(\frac{Lc_u^2 \eta^2}{2} - \frac{c_l \eta}{2}\right)\mathbb{E}\left\|\mathbf{m}^k\right\|^2 + \frac{2Lc_1 c_u c_\infty \eta^2 d}{\beta} \\
&\le \mathbb{E}f(\boldsymbol{\theta}^k) + \frac{c_u \eta}{2}\mathbb{E}\left\|\nabla f(\tilde{\boldsymbol{\theta}}^k) - \mathbf{m}^k\right\|^2 - \frac{c_l \eta}{2}\mathbb{E}\left\|\nabla f(\tilde{\boldsymbol{\theta}}^k)\right\|^2 \\
&\quad - \frac{c_l \eta}{4}\mathbb{E}\left\|\mathbf{m}^k\right\|^2 + \frac{2Lc_1 c_u c_\infty \eta^2 d}{\beta}.
\end{aligned}
\quad (54)
$$

For the first inequality we use (52). For the second inequality we use the upper and lower bound of $\boldsymbol{\eta}$. For the third inequality we set $\eta \le \frac{c_l}{2Lc_u^2}$. Then based on Lemma C.4 we have

$$\mathbb{E}\left\|\mathbf{m}^k - \nabla f(\tilde{\boldsymbol{\theta}}^k)\right\|^2 \le (1 - \beta_1)\mathbb{E}\left\|\mathbf{m}^{k-1} - \nabla f(\tilde{\boldsymbol{\theta}}^{k-1})\right\|^2 + \frac{(1 - \beta_1)^2 L^2}{\beta_1}\mathbb{E}\left\|\tilde{\boldsymbol{\theta}}^{k-1} - \tilde{\boldsymbol{\theta}}^k\right\|^2 + \frac{\beta_1^2 \sigma^2}{N}. \quad (55)$$

For the second factor we have

$$\mathbb{E}\left\|\tilde{\boldsymbol{\theta}}^{k-1} - \tilde{\boldsymbol{\theta}}^k\right\|^2 \le 3\left(\mathbb{E}\left\|\boldsymbol{\theta}^{k-1} - \boldsymbol{\theta}^k\right\|^2 + \mathbb{E}\left\|\tilde{\boldsymbol{\theta}}^k - \boldsymbol{\theta}^k\right\|^2 + \mathbb{E}\left\|\tilde{\boldsymbol{\theta}}^{k-1} - \boldsymbol{\theta}^{k-1}\right\|^2\right). \quad (56)$$

Thus, we have

$$
\mathbb{E}\left\|\mathbf{m}^k - \nabla f(\tilde{\boldsymbol{\theta}}^k)\right\|^2 \leq (1-\beta_1)\mathbb{E}\left\|\mathbf{m}^{k-1} - \nabla f(\tilde{\boldsymbol{\theta}}^{k-1})\right\|^2 + \frac{3c_u^2\eta^2(1-\beta_1)^2L^2}{\beta_1}\mathbb{E}\|\mathbf{m}^{k-1}\|^2
$$
$$
+ \frac{\beta_1^2\sigma^2}{N} + \frac{24L^2c_1^2c_u^2\eta^2 d}{\beta^2}.
$$
(57)

By adding equation (54) at $\boldsymbol{\theta}^{k+1}$ and $a\times$ equation (57) at $\boldsymbol{\theta}^{k+1}$, we have

$$
\mathbb{E}f(\boldsymbol{\theta}^{k+1}) + a\mathbb{E}\left\|\mathbf{m}^{k+1} - \nabla f(\tilde{\boldsymbol{\theta}}^{k+1})\right\|^2 \leq \mathbb{E}f(\boldsymbol{\theta}^k) + \left(\frac{c_u\eta}{2} + a(1-\beta_1)\right)\mathbb{E}\left\|\nabla f(\tilde{\boldsymbol{\theta}}^k) - \mathbf{m}^k\right\|^2
$$
$$
- \frac{c_l\eta}{2}\mathbb{E}\left\|\nabla f(\tilde{\boldsymbol{\theta}}^k)\right\|^2 - \left(\frac{c_l\eta}{4} - \frac{3ac_u^2\eta^2(1-\beta_1)^2L^2}{\beta_1}\right)\mathbb{E}\left\|\mathbf{m}^k\right\|^2
$$
$$
+ \frac{a\beta_1^2\sigma^2}{N} + \frac{2Lc_1c_uc_\infty\eta^2 d}{\beta} + \frac{24aL^2c_1^2c_u^2\eta^2 d}{\beta^2}.
$$
(58)

Letting $a = \frac{\eta c_u}{\beta_1}$ and $G(\boldsymbol{\theta}^k) = \mathbb{E}f(\boldsymbol{\theta}^k) + \frac{\eta c_u}{\beta_1}\mathbb{E}\left\|\mathbf{m}^{k+1} - \nabla f(\tilde{\boldsymbol{\theta}}^{k+1})\right\|^2$, we have

$$
G(\boldsymbol{\theta}^{k+1}) \leq G(\boldsymbol{\theta}^k) - \frac{c_l\eta}{2}\mathbb{E}\left\|\nabla f(\tilde{\boldsymbol{\theta}}^k)\right\|^2 - \left(\frac{c_l\eta}{4} - \frac{3c_u^3\eta^3(1-\beta_1)^2L^2}{\beta_1^2}\right)\mathbb{E}\left\|\mathbf{m}^k\right\|^2
$$
$$
+ \frac{c_u\eta\beta_1\sigma^2}{N} + \frac{2Lc_1c_uc_\infty\eta^2 d}{\beta} + \frac{24L^2c_1^2c_u^3\eta^3 d}{\beta_1\beta^2}
$$
$$
\leq G(\boldsymbol{\theta}^k) - \frac{c_l\eta}{2}\mathbb{E}\left\|\nabla f(\tilde{\boldsymbol{\theta}}^k)\right\|^2 - \frac{c_l\eta}{8}\mathbb{E}\left\|\mathbf{m}^k\right\|^2
$$
$$
+ \frac{c_u\eta\beta_1\sigma^2}{N} + \frac{2Lc_1c_uc_\infty\eta^2 d}{\beta} + \frac{24L^2c_1^2c_u^3\eta^3 d}{\beta_1\beta^2},
$$
(59)

where we set $\eta \leq \frac{\beta_1 c_l^{0.5}}{5c_u^{1.5}(1-\beta_1)L}$. Then we sum the inequality above from $k=0$ to $K-1$ and obtain

$$
\frac{1}{K}\sum_{k=0}^{K-1}\mathbb{E}\left[\left\|\nabla f(\tilde{\boldsymbol{\theta}}^k)\right\|^2 + \frac{1}{4}\left\|\mathbf{m}^k\right\|^2\right] \leq \frac{2(G(\boldsymbol{\theta}^0) - G(\boldsymbol{\theta}^K))}{\eta c_l K} + \frac{2c_u\beta_1\sigma^2}{c_l N} + \frac{4Lc_1c_uc_\infty\eta d}{c_l\beta}
$$
$$
+ \frac{48L^2c_1^2c_u^3\eta^2 d}{c_l\beta_1\beta^2}.
$$
(60)

Considering that

$$
G(\boldsymbol{\theta}^0) - G(\boldsymbol{\theta}^K) \leq \mathbb{E}f(\boldsymbol{\theta}^0) - \mathbb{E}f(\boldsymbol{\theta}^K) + \frac{\eta c_u}{\beta_1}\mathbb{E}\left\|\mathbf{m}^0 - \nabla f(\tilde{\boldsymbol{\theta}}^0)\right\|^2
$$
$$
\leq \mathbb{E}f(\boldsymbol{\theta}^0) - \mathbb{E}f(\boldsymbol{\theta}^*) + \frac{\eta c_u\sigma^2}{\beta_1 N},
$$
(61)

and letting $\mathbb{E}f(\boldsymbol{\theta}^0) - \mathbb{E}f(\boldsymbol{\theta}^*) = \Delta$, we have

$$
\frac{1}{K}\sum_{k=0}^{K-1}\mathbb{E}\left[\left\|\nabla f(\tilde{\boldsymbol{\theta}}^k)\right\|^2 + \frac{1}{4}\left\|\mathbf{m}^k\right\|^2\right] \leq \frac{2\Delta}{\eta c_l K} + \frac{2c_u\sigma^2}{c_l K\beta_1 N} + \frac{2c_u\beta_1\sigma^2}{c_l N} + \frac{4Lc_1c_uc_\infty\eta d}{c_l\beta}
$$
$$
+ \frac{48L^2c_1^2c_u^3\eta^2 d}{c_l\beta_1\beta^2}.
$$
(62)

Finally setting $K = \Omega\left(d\epsilon^{-4}\right)$, $\eta = \mathcal{O}\left(d^{-1}\epsilon^2\right)$ and $\beta_1 = \mathcal{O}\left(d^{-1}\epsilon^2\right)$, we have

$$
\frac{1}{K}\sum_{k=0}^{K-1}\mathbb{E}\left[\left\|\nabla f(\tilde{\boldsymbol{\theta}}^k)\right\|^2 + \frac{1}{4}\left\|\mathbf{m}^k\right\|^2\right] \leq \mathcal{O}\left(\epsilon^2\right).
$$
(63)

We end the proof of Theorem 5.7. $\qquad\square$

# D  LOW-RANK PROPERTY OF GRADIENT IN TRAINING DYNAMIC

In this section, we give an simple theoretical analysis to demonstrate that the gradient of project-up layers of MLP in Transformer-based models becomes low-rank with training time. Our analysis is based on recent study on the framework of Transformer dynamic, JoMA [50], and fix an error in related study [67].

We follow the definitions and assumptions of Transformer in JoMA [50], and compute the dynamic of project-up layer in Transformer MLP as below.

**Lemma D.1.** *Suppose the embedding matrix $\mathbf{U} \in \mathbb{R}^{m \times M}$ is fixed, where $M$ represents the vocabulary size. The active function is linear and the back propagation gradient is stationary [50]. The weight matrix is $\mathbf{W} \in \mathbb{R}^{m \times n}$. Let $\Delta = [\mathbf{\Delta}_1, \cdots, \mathbf{\Delta}_n]$, where $\mathbf{\Delta}_j = \mathbb{E}_q[g_j \mathbf{x}] \in \mathbb{R}^M$. $g_j$ is the back propagated gradient of hidden node $j$ in MLP, $\mathbb{E}_q$ represents the conditional expectation given the query $q$, and $\mathbf{x}$ represents the token distribution in the previous layer, for example, the activation output of the previous layer. Following the same definitions and assumptions in JoMA [50], the project-up matrix $\mathbf{V} = \mathbf{U}^\top \mathbf{W} \in \mathbb{R}^{M \times n}$ satisfis*

$$\dot{\mathbf{V}} = \frac{1}{A} \exp \left( \frac{\mathbf{V} \circ \mathbf{V}}{2} \right) \circ \Delta, \tag{64}$$

*where $A$ is the normalization of softmax.*

*Proof.* Let $\mathbf{V} = [\mathbf{v}_1, \cdots, \mathbf{v}_n]$ and $\mathbf{W} = [\mathbf{w}_1, \cdots, \mathbf{w}_j]$. Following the Theorem 2 in study of JoMA framework [50], for each hidden node $j$, $\mathbf{v}_j = \mathbf{U}^\top \mathbf{w}_j$ satisfies

$$\dot{\mathbf{v}}_j = \frac{1}{A} \mathbf{\Delta}_j \circ \exp \left( \mathbf{v}_j^2 / 2 \right), \tag{65}$$

where $\mathbf{v}_j^2 = \mathbf{v}_j \circ \mathbf{v}_j$ is the element-wise product of vector. Then we notice that Lemma D.1 is the matrix form of equation (65). $\qquad \square$

With Lemma D.1, we analyse the training dynamic of $\mathbf{V}$. Row-wisely write $\mathbf{V}$ as $\mathbf{V} = \left[ \mathbf{u}_1^\top, \cdots, \mathbf{u}_M^\top \right]^\top$ and $\Delta$ as $\Delta = \left[ \boldsymbol{\delta}_1^\top, \cdots, \boldsymbol{\delta}_M^\top \right]^\top$. Based on Lemma D.1, for $j \in [M]$, we have

$$\dot{\mathbf{u}}_j = \frac{1}{A} \exp \left( \mathbf{u}_j^2 / 2 \right) \circ \boldsymbol{\delta}_j = \frac{1}{A} \text{diag} \left( \exp \left( \mathbf{u}_j^2 / 2 \right) \right) \boldsymbol{\delta}_j \tag{66}$$

Based on the assumption that the back propagation gradient is stationary and $\mathbf{x}$ is the output of previous layer, which means it is given. Thus the $\mathbf{\Delta}_j$ is constant, and the direction of $\mathbf{u}_j$ is controlled by a diagonal matrix. Let $\dot{\mathbf{u}}_j = \mathbf{D}_j(t) \boldsymbol{\delta}_j$, where $\mathbf{D}_j(t)$ is a diagonal matrix with initialization $\mathbf{D}_j(0) = \mathbf{0}$. Then we have

$$\dot{\mathbf{D}}_j(t) = \frac{1}{A} \text{diag} \left( \exp \left( \frac{\mathbf{D}_j^2(t) \boldsymbol{\delta}_j^2}{2} \right) \right). \tag{67}$$

Let $a_j^2 = \max_{i \in [n]} \delta_{ji}^2$ and $b_j^2 = \min_{i \in [n]} \delta_{ji}^2$. Then we have

$$\frac{1}{A} \text{diag} \left( \exp \left( \frac{\mathbf{D}_j^2(t) b_j^2}{2} \right) \right) \preceq \dot{\mathbf{D}}_j(t) \preceq \frac{1}{A} \text{diag} \left( \exp \left( \frac{\mathbf{D}_j^2(t) a_j^2}{2} \right) \right). \tag{68}$$

Element-wisely analyze the diagnoal matrix $\mathbf{D}_j(t) = \text{diag} \left( d_{j1}(t), \cdots, d_{jn}(t) \right)$. For $i \in [n]$, we have

$$\frac{1}{A} \exp \left( \frac{d_{ji}^2(t) b_j^2}{2} \right) \leq \dot{d}_{ji} \leq \frac{1}{A} \exp \left( \frac{d_{ji}^2(t) a_j^2}{2} \right). \tag{69}$$

For dynamic like $\dot{x} = C e^{\beta^2 x^2}$, we have

$$x(t) = \frac{1}{\beta} \text{erf}^{-1} \left( \frac{2 \beta C}{\sqrt{\pi}} t \right), \tag{70}$$

Table 4: Comparison of SEPARATE with previous common gradient compression algorithms in standard application metrics, including gradient complexity , real communication time, memory cost, ring-based communication support, and sharding support. Here, $\Psi$ represents the scale of model parameters, $N$ represents the number of nodes in distributed cluster, $B$ is the communication bandwidth (bytes/s), $r$ is the dimension of low-rank subspace of gradient to PowerSGD, and $m$ is the random Gaussian samples in SEPARATE in Algorithm 1. Here we only consider the order of $\epsilon$ in gradient complexity, ignoring others especially the dimension of parameters $d$. We discuss the order of $d$ in detail in Section 5. We consider the mixed-precision setting for memory overhead.

| Methods | Gradient Complexity | Communication Cost | Memory Cost | Ring-based Comm. | Gradient Sharding |
|---|---|---|---|---|---|
| SGD | $\mathcal{O}\left(\epsilon^{-4}\right)$ | $4\Psi(N-1)/(BN)$ | $2\Psi + 6\Psi/N$ | ✓ | ✓ |
| Adam [27] | $\mathcal{O}\left(\epsilon^{-4}\right)$ | $4\Psi(N-1)/(BN)$ | $2\Psi + 14\Psi/N$ | ✓ | ✓ |
| 1-bit Adam [49] | $\mathcal{O}\left(\epsilon^{-4}\right)$ | $0.625\Psi N/B$ | $18\Psi$ | ✗ | ✗ |
| 1-bit LAMB [29] | $\mathcal{O}\left(\epsilon^{-4}\right)$ | $0.625\Psi N/B$ | $22\Psi$ | ✗ | ✗ |
| PowerSGD [55] | ✗ | $4r\sqrt{\Psi}(N-1)/(BN)$ | $14\Psi + 2r\sqrt{\Psi}$ | ✓ | ✓ |
| SEPARATE-SGD | $\mathcal{O}\left(\epsilon^{-4}\right)$ | $4m(N-1)/(BN)$ | $4\Psi + 6\Psi/N$ | ✓ | ✓ |
| SEPARATE-Adam | $\mathcal{O}\left(\epsilon^{-4}\right)$ | $4m(N-1)/(BN)$ | $4\Psi + 14\Psi/N$ | ✓ | ✓ |

where $\text{erf}(x)$ is Gaussian error function. Then we have

$$\frac{\sqrt{2}}{b_j}\text{erf}^{-1}\left(\sqrt{\frac{2}{\pi}}\frac{b_j}{A}t\right) \leq d_{ji}(t) \leq \frac{\sqrt{2}}{a_j}\text{erf}^{-1}\left(\sqrt{\frac{2}{\pi}}\frac{a_j}{A}t\right). \tag{71}$$

Since the bound is independent of $i$, for each $j$ we have

$$\frac{\sqrt{2}}{b_j}\text{erf}^{-1}\left(\sqrt{\frac{2}{\pi}}\frac{b_j}{A}t\right)\mathbf{I} \preceq \mathbf{D}_j(t) \preceq \frac{\sqrt{2}}{a_j}\text{erf}^{-1}\left(\sqrt{\frac{2}{\pi}}\frac{a_j}{A}t\right)\mathbf{I}. \tag{72}$$

It demonstrates that for each line of $\mathbf{V}$, the direction of $\mathbf{u}_j$ is near to the direction of $\boldsymbol{\delta}_j$ with error dependent to $a_j$ and $b_j$. Next we show that the change of direction from one line to another is very different resulting in low-rank property of $\mathbf{V}$. Let

$$h(t,a) = \frac{\sqrt{2}}{a}\text{erf}^{-1}\left(\sqrt{\frac{2}{\pi}}\frac{a}{A}t\right). \tag{73}$$

Then we have $h(t,b_j)\mathbf{I} \preceq \mathbf{D}_j(t) \preceq h(t,a_j)\mathbf{I}$. Considering $\lim_{t \to \sqrt{\pi}A/\sqrt{2}a} h(t,a) \to +\infty$, let $j^* = \arg\max_j b_j$ be the row with the largest entry of $b_j$. Then if $b_{j^*} \geq a_j$ for all $j \neq j^*$, when $t \to t^* = \sqrt{\pi}A/\sqrt{2}b_{j^*}$, we have $\mathbf{D}_{j^*}(t) \succeq h(t,b_{j^*})\mathbf{I} \to +\infty$ but $\mathbf{D}_j(t) \preceq h(t,a_j)\mathbf{I}$ is still finite because $t' = \sqrt{\pi}A/\sqrt{2}a_j \geq t^*$.

Such analysis demonstrates that the magnitude of each row of $\mathbf{V}$ changes drastically. The dominant direction of $\mathbf{V}$ will become extremely large when others are still in a small range over time. $\mathbf{V}$ will become near rank-1 over time and $\dot{\mathbf{V}}$ has lower rank because $\dot{\mathbf{D}}$ have $\mathbf{D}$ on exponents. Such analysis is consistent with what we observed in Figure 1 in Section 3.

## E  EXPEIMENT DETAILS

### E.1  MODEL CONFIGURATIONS

We pre-train the GPT-2-345M [40] model on OpenWebtext dataset [17] of 10B tokens from scratch. We show the results in Figure 2. Second, we compare the SEPARATE with several representative baselines to fine-tune the LLAMA2-7B model and evaluate on downstream tasks. The baselines

Table 5: Compare SEPARATE with LoRA. Suppose that the weight matrix $\mathbf{W} \in \mathbb{R}^{m \times n}$, the low-rank reparameterized size of LoRA is $r$, and the compression ratio of SEPARATE is $k$.

|  | SEPARATE | LoRA |
|---|---|---|
| Weight | $mn$ | $mn + nr + mr$ |
| Grad Comm. | $mn/k$ | $nr + mr$ |
| Pre-train | ✓ | ✗ |
| Fine-tune | ✓ | ✓ |

Table 6: Performance comparison with LoRA. We fine-tune LLAMA2-7B on the alpaca-gpt4 dataset and evaluate the performance on downstream tasks, including commonsense reasoning, world knowledge, math, and code.

| Method | TriQA | GSM8K | MBPP | NQ | WinoG | Arc-e | Arc-c | PIQA | Avg. |
|---|---|---|---|---|---|---|---|---|---|
| Adam | 49.39 | 15.69 | 19.40 | 3.07 | 46.09 | 72.31 | **53.90** | **57.07** | 39.62 |
| Adam-LoRA | **62.08** | 16.53 | 16.80 | **5.15** | **49.57** | 49.21 | 37.97 | 52.88 | 36.27 |
| SEPARATE | 57.18 | **20.17** | **21.40** | 4.18 | 49.25 | **72.66** | 49.83 | 52.99 | **40.96** |

include typical low-rank optimizers with error feedback technique like PowerSGD [55], representative low-bit optimizers like 1-bit Adam [49], and recent quantization efficient training strategy ZeRO++ [56]. For fairness, we use SEPARATE-based Adam for comparison. In this section, we propose the experiment details including model and hyper-parameters settings. Without special emphasis all experiments are under bfloat16 precision.

**GPT-2.** We conducted the training of GPT-2-345M model [40] from scratch on the OpenWebtext [17] dataset on $8\times$ NVIDIA 3090 24G GPUs cluster. We ran a total of 50000 iterations and processed a total of 10B tokens. The experiment was configured with PyTorch DDP training framework without model and pipeline parallelism to simplify the experimental setting and fit existing devices. Our global batch size was $8\times 512$. We set the learning rate at 6.0e-4 with the cosine decay down to the minimum 6.0e-5 after 2000 iterations of warm-up. We also used the gradient accumulation and set the gradient accumulation step at 4. We used the global gradient norm clipping of 1 and set the AdamW with $\beta_1 = 0.9$, $\beta_2 = 0.95$ and weight decay as $0.1$.

**LLAMA2-7B.** We conducted the fine-tuning of LLAMA2-7B [51] model on alpaca-gpt4 dataset [39] within the PyTorch FSDP [68] framework. We followed protocols from LLAMA-Accessory [66] and fine-tuned LLAMA2-7B for three epochs on $8\times$ NVIDIA A6000 48G GPUs cluster. To fit the devices we set the data parallelism at 4 and model parallelism at 2 without pipeline parallelism. We set the global batch size at 32, the learning rate of 2e-5, gradient clipping of 2 and gradient accumulation step at 8.

### E.2 BENCHMARKS

Following the previous work [66], we conducted evaluation on chosen benchmarks on the popular OpenCompass [12] platform. We chose the benchmarks including commonsense reasoning, world knowledge, math and code as follows:

- Commonsense Reasoning: Hellaswag [65], Winogrande [45], ARC-Easy [10], ARC-Challenge [10] and PIQA [7].
- World Knowledge: NaturalQuestions [28] and TriviaQA [26].
- Math: GSM8K [11].
- Code: MBPP [4].

### E.3 APPLICATION TO TRAINING PROCESS AS A PLUG-IN

SEPARATE can be easily applied to real large-scale model training tasks. SEPARATE only needs small amount of extra computation costs, which can be formulated as tensor multiplication form that

---

**Algorithm 3** G-SEPARATE: General Version of Simple Low-rank Projection

---

**Require:** Initialization model parameters with $L$ layers, $N$ nodes, layer-wise communication ratio $\{m_l\}_{l=1}^{L}$, layer-wise $\{\beta_l\}_{l=1}^{L}, \beta_l = 0.95, \forall l \in [L]$, error reset frequency $T_e$, adaptive update frequency $T_a$, a common Gaussian random number generator, initialize $\mathbf{e}^0 \in \mathcal{B}(\mathbf{0}, c_1)$

**while** $k \leq K$ **do**

STEP 1. In each node $n$ compute stochastic gradient $\mathbf{g}_{n,l}^k$ and $\mathbf{h}_{n,l}^k = \mathbf{g}_{n,l}^k + \mathbf{e}_{n,l}^k$;

STEP 2. Generate fresh i.i.d. common random Gaussian vectors $\boldsymbol{\xi}_1, \cdots, \boldsymbol{\xi}_{m_l^k} \sim N(0, \mathbf{I}_d)$

and compute $[p_{1,n,l}, \cdots, p_{m_l^k,n,l}]$ with $p_{i,n,l} = \langle \mathbf{h}_{n,l}^k, \boldsymbol{\xi}_i \rangle$ as the low-dimension projection of $\mathbf{h}_{n,l}^k$;

STEP 3. Do all-reduce and obtain global projected gradient $[\tilde{p}_{1,l}, \cdots, \tilde{p}_{m_l^k,l}]$;

STEP 4. Compute $\tilde{\mathbf{h}}_{n,l}^k = \frac{1}{m_l^k} \sum_{i=1}^{m_l^k} \tilde{p}_{i,l} \cdot \boldsymbol{\xi}_i$ and use $\tilde{\mathbf{h}}_{n,l}^k$ for model weight update in node $n$;

STEP 5. Update error

$\mathbf{e}_{n,l}^{k+1} = (1 - \beta_l^k)\mathbf{e}_{n,l}^k + \beta_l^k(\mathbf{h}_{n,l}^k - \tilde{\mathbf{h}}_{n,l}^k)$ if $k\%T_e \neq 0$,

$\mathbf{e}_{n,l}^{k+1} = 0$ if $k\%T_e = 0$ (error reset);

STEP 6. Layer-wisely update compression ratio and $\beta_l$

$m_l^{k+1} = \text{int}\left(1 + m_l^k \cdot \left(1 + \frac{\langle \tilde{\mathbf{h}}_{n,l}^k, \mathbf{g}_{n,l}^k \rangle}{\|\tilde{\mathbf{h}}_{n,l}^k\| \cdot \|\mathbf{g}_{n,l}^k\|}\right)\right)$,

$\beta_l^{k+1} = \max\left\{\min\left\{\beta_l^k \cdot \left(1 + \frac{\langle \tilde{\mathbf{h}}_{n,l}^k, \mathbf{g}_{n,l}^k \rangle}{\|\tilde{\mathbf{h}}_{n,l}^k\| \cdot \|\mathbf{g}_{n,l}^k\|}\right), 0.99\right\}, 0.90\right\}$ if $k\%T_a = 0$,

$m_l^{k+1} = m_l^k$,

$\beta_l^{k+1} = \beta_l^k$, if $k\%T_a \neq 0$;

**end while**

---

is well supported by modern computational device architectures like CUDA [35]. By using **hook** mechanism in PyTorch, we can compress the latter layer's gradient and do communication while the optimizer computes the former layer's gradient in backpropagation process. Thus we properly arrange the overlapped computation and communication order, the time cost of compression can be hidden in the backpropagation computation, and SEPARATE can realize faster training time. Moreover, SEPARATE can be applied to ring-based or tree-based communication. Different from quantization methods, SEPARATE is not troubled by numerical anomalies like overflow or underflow, so it well supports reduce-scatter operation. SEPARATE also supports sharding strategy which is commonly used for training large language models [41]. Even though the gradient is separated into different nodes, the linearity of random projection ensures the correctness of SEPARATE's results. We propose an example that how to use SEPARATE as a plug-in by hook mechanism in Algorithm 2.

Besides, SEPARATE brings extra memory costs for training because of the store of error and random variables. Setting up a random variable buffer, we can reduce the memory costs by reusing this part of memory to store the stream of random variables. In summary, from the results in Table 4, it is evident that SEPARATE demonstrates superior properties compared to other methods.

### E.4   ADDITIONAL EXPERIMENTS

Parameter-efficient fine-tuning (PEFT) techniques accelerate the application of pre-trained language models to different downstream tasks without the need to fine-tune all of the model's parameters [14]. Among them, the popular Low-Rank Adaptation (LoRA [22]) reparameterizes the weight matrix to low-rank approximations and fine-tuning the reparameterized ones. We compare SEPARATE with popular LoRA on scopes of application and performance on downstream tasks in Table 5 and Table 6. The results demonstrate that SEPARATE has better versatility and effect than LoRA.

### E.5   GENERAL VERSION OF SIMPLE LOW-RANK PROJECTION

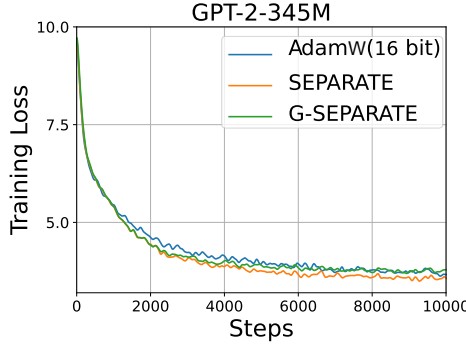

Figure 4: Loss curve of vanilla Adam, SEP-ARATE with compression ratio 8, and G-SEPARATE on GPT-2-345M trained with 10B tokens OpenWebtext dataset.

In this section, we study how to adaptively select the compression ratio and $\beta$ for moving average error feedback. We expect to dynamically adjust these hyperparameters by combining the characteristics of the parameters at each layer of the model and the training dynamic. Our research focuses on two aspects. First, the adaptive strategy needs to remain simple and efficient to ensure the overall training wall-clock time is reduced. Thus, we still do not consider the strategies that require periodic heavy computations, such as periodic SVD. Second, considering the robustness of our method shown in Appendix A, we think the dynamic adjustment of hyperparameters should not be too sharp to avoid instability in training. Under this consideration, we get the general version of SEPARATE in Algorithm 3.

We layer-wisely compute the cosine similarity of the gradient estimate and the gradient itself frequently to evaluate the estimate accuracy, and update the compression ratio and $\beta$ in moving average error feedback dynamically. As shown in Algorithm 3 STEP 6, if the estimate is accurate, we try to use a more aggressive compression ratio and a larger $\beta$ in the nest stage, and vice versa. All steps in Algorithm 3 are layer-wise, and we use the subscript $l$ to represent the operation on the $l$-th layer. The initialization of $m_l$ can be arbitrary such as 16, 32 or 64. The adaptive mechanism can adjust it to the proper region of the corresponding layer through several dynamic update processes.

We pre-train GPT-2-345M on 10B tokens OpenWebtext dataset from scratch to verify the effectiveness of G-SEPARATE. We follow the same hyperparameter setting of our pre-training experiment in Appendix E.1. For G-SEPARATE, we set the adaptive update frequency $T_a = 2000$ to ensure the stability of training. The results shown in Figure 4 indicate that the adaptive extension more slightly fits the baseline, but shares the similar performance of the original algorithm in total.

