# OpenReview forum: "SEPARATE: A Simple Low-rank Projection for Gradient Compression in Modern Large-scale Model Training Process"
_ICLR.cc/2025/Conference — ICLR 2025 Poster_

### Official Review · Reviewer_nVYQ · 2024-10-16

**Soundness:** 3
**Presentation:** 2
**Contribution:** 2
**Rating:** 6
**Confidence:** 3

**Summary:**

The paper proposes a random projection compression scheme for speeding up gradients' all-reduce operations in distributed training setting (e.g. DDP) via random Gaussian projection. By relying on the unbiased down project and up project and that the variance is bounded, convergence of stochastic optimizer like SGD and Adam are still guaranteed, albeit the bounds suffer a bit from the increased noise from the compression scheme. The authors provide experiments showing that the compression schemes speed up training compared to baselines while does not suffer too much performance degredation.

**Strengths:**

• Experimental speedup seems useful.

**Weaknesses:**

• Contributions: The compression scheme is not too new and have been recently explored in a variety of papers like GaLore and Flora (also uses dense Gaussian projection). While the focus of this paper is on communication, I do not see any new idea since the main ideas in GaLore and Flora are also to compress the gradients via some form of projection and then perform optimization in the compressed space before projecting it back up for the update. Extending the ideas from GaLore and Flora to the distributed setting seems straightforward to me. Furthermore, the convergence guarantees are just simple adaptation of well-known proofs i.e. convergence for unbiased and light-tailed noise (bounded variance).

• The moving average error feedback section is not well written. The authors make several claims that are not obvious to me. For example, “[due to the] instability and discontinuity of random projection, the compression error may fluctuate when the random directions are far from the dominant directions of Hessian or the random directions change acutely” is not obvious since the Hessian describes properties of the optimization landscape and the compression scheme impacts (primarily) the direction of the gradient. I think the authors should rewrite the section to compare with the previous error-feedback work and argue better for why their use of EMA and reset are better.

• Experiments and reproducibility concerns: The authors should disclose the tuning effort and hyperparameters for all methods being compared. I couldn't find how the experiments are conducted to determine the learning rate and what parameters are tuned for what amount. Providing code and reproduction steps should be standard. An improvement is not significant if one method is tuned (i.e. given a lot more compute) significantly more than others. Also, experimental improvement for the error feedback seems insignificant and noise prone. From Table 2, I couldn't tell if the error-feedback helps or not.

References:
• Zhao, Jiawei, et al. "Galore: Memory-efficient llm training by gradient low-rank projection." arXiv preprint arXiv:2403.03507 (2024).

• Hao, Yongchang, Yanshuai Cao, and Lili Mou. "Flora: Low-Rank Adapters Are Secretly Gradient Compressors." arXiv preprint arXiv:2402.03293 (2024).

**Questions:**

• I'm sure I understand the authors' connection between the gradient being low rank and the eigenvalues of the Hessian. For gradient compression, low rank could imply that there is some low-error decomposition but I'm not sure what the top-heavy eigenvalues have to do with gradient compression?

• In defining the error feedback, what is the domain for the argmin of e? Isn't that just the orthogonal component of the gradient relative to the random projection subspace? Also, how do you justify the claim “taking moving average of the historical error maintains the stability and variance of accumulated error?” I am not familiar with

• The authors should consider other sketching scheme like fast-JL and importance sampling scheme that are much faster than dense Gaussian projection.

• Theory requires fresh i.i.d. Gaussian matrix to be generated across all nodes identically every round to ensure the compression and decompression is unbiased. What is the additional cost to communicate this?

---

> ### Author Response · Authors · 2024-11-24
>
> # Resopnse to Reviewer nVYQ
>
> We sincerely thank the reviewer nVYQ for the valuable and constructive comments. We have updated the submitted paper and try to clarify your concerns. We start with an important discussion to address your concerns about our method and theory in total. Then we clarify the weaknesses and questions one by one.
>
>
> ## Important Discussion
>
> We begin with expressing our gratitude for your insightful observation regarding the application of related techniques to diverse tasks, exemplified by GaLore and Flora. However, what we need to clarify is that communication-efficient training has specific challenges that make our "simple but efficient" method necessary.
>
> First, it has indeed come to our attention during our investigation that updating the optimizer state within a low-dimensional subspace is perceived as a strategy to mitigate memory overhead. However, it is imperative to clarify that these methods cannot be directly applied to the reduction of communication overhead.  The reduction of memory and the mitigation of communication overhead need to be recognized as distinct challenges for efficient training.  GaLore and our approach each address different sides of efficient training.  While there exists the potential for a synergistic combination of these two, the additional computational burden is deemed more intolerable for communication reduction. Specifically, for memory reduction, the methods introduce some frequent heavy computation (like SVD) to maintain the accuracy of the estimate, but for communication reduction, to reduce the wall-clock time, lightweight computation is necessary. The primary goal of communication cost reduction is to reduce the overall wall-clock training time.  To achieve this, our "simple but efficient" method is currently the only solution.  Under frequent communication rounds, the computationally intensive nature of SVD and analogous operations becomes prohibitive.  A more detailed discussion of this topic has been presented in updated Appendix A.
>
> Second, in the subsequent discourse, we delve into a more detailed sketch of our theoretical analysis. We start with the variance analysis shown in **Lemma 4.2**. This lemma serves as a foundation in our theoretical analysis framework, elucidating that the variance inherent in our estimate can be effectively bounded by the **trace** of the Hessian, as we discuss in **Remark 4.3**. The observation of the "top-heavy" of the Hessian is significant. It is a pivotal property that ensures a **bounded trace**, as rigorously defined in **Assumption 5.4**. This boundedness of the trace is a critical factor in substantiating the bounded variance, which in turn underpins the main **Theorem 5.5**. Within this theorem, we achieve a convergence rate of $\Omega(d^{1/2}\epsilon^{-4})$ in the standard setting. The result transcends a mere adaptation of existing methods. It represents a significant **improvement**. This improvement is particularly noteworthy when compared with the conventional convergence rate of gradient compression techniques $\mathcal{O}(d\epsilon^{-4})$, as we discuss in **Remark 5.6**. It signifies a substantial advancement in communication-efficient optimization algorithms.   This improvement is a testament to the efficacy of our approach in mitigating the computational complexities associated with high-dimensional data, thereby offering a more streamlined path to convergence. This work is a cornerstone of our contribution, offering a deeper insight into the interplay between the Hessian's structure, the trace's boundedness, and the consequent impact on the convergence of optimizers.

---

> > ### Comment · Reviewer_nVYQ · 2024-11-24
> >
> > Dear authors,
> > Thank you for your thorough response. I appreciate the clarifications and revisions. I have several followup questions:
> >
> > - Is the dimensional improvement in remark 5.6 due to a better analysis or algorithmic innovation? It would be helpful if you could highlight the key point.
> >
> > - Can you help me understand how to get equation (46) on line 992-993? Is it due to assumption 5.4?  Also regarding assumption 5.4, don't we already have the Hessian is $\le L\cdot I$ due to the L-Lipschitzness of the gradients?
> >
> > - Regarding the experimental details, I am hoping to learn more about how each baseline like Adam and PowerSGD is tuned i.e. what is the search space (if any) for the hyperparameters like learning rate and batch size? I guess I am surprise at why random projections of gradients could improve over Adam. Especially since the gradients are mostly low-rank, wouldn't random projections most likely to project the gradient to some space with small singular values? If we tune Adam more, esp the base learning rate, could we get a better result?
> >
> > - Perhaps the authors could add comparison with the following work that also uses sketching to reduce communication efficiency [Rothchild et al. 2020] FetchSGD: Communication-Efficient Federated Learning with Sketching. The CountSketch seems faster than dense projection.
> >
> > - Finally, regarding the fast-JL numbers, perhaps the implementation details matter (such as kernel fusions) because in my experience, the sparse sketching schemes like fastJL and countsketch are typically much faster than dense gaussian if implemented correctly (due to fast FFT implementation, small number of random samples which can be time consuming, etc.). However, this is just an extension and not important here. Just some extension that I can think of. Thank you for trying fastJL out though.

---

> > > ### Author Response · Authors · 2024-11-25
> > >
> > > # Resopnse to Reviewer nVYQ
> > >
> > > We appreciate your patience in reading our responses and giving new questions and suggestions. We try to clarify your concerns in the following three parts.
> > >
> > > ## Part I. Clarification on Algorithm and Theory
> > >
> > > ### Q1: Is the dimensional improvement in remark 5.6 due to a better analysis or algorithmic innovation?
> > > We start with the discussion that **the dimension improvement in Remark 5.6 is due to our algorithm innovation.**  We give an example under the **$\mu$-strongly convex setting** as below to illustrate how our algorithm achieves the dimension improvement intuitively. The example is
> > >
> > > $\min_{\mathbf{\theta} \in R^d} f(\mathbf{\theta}) = \frac{1}{2}\left( L(\theta_1 -\theta_1^*)^2 + \sum_{i=2}^d \mu(\theta_i - \theta_i^*)^2 \right)$,
> > >
> > > where $\mathbf{\theta} = [\theta_1,\cdots,\theta_d]$, $\mathbf{\theta}^* = [\theta_1^*,\cdots,\theta_d^*]$. $L$ and $\mu$ satisfy $(d-1)\mu \approx L$, indicating that the Hessian of $f$ (we use $\mathbf{A}$ to represent) is "top-heavy", and the trace of Hessian is bounded as $tr(\mathbf{A}) \approx 2L \ll dL$. For this example, if we use gradient descent to find the optimal solution, we can write the gradient as
> > >
> > > $\nabla f(\mathbf{\theta}) = \left(L(\theta_1 - \theta_1^*), \mu(\theta_2 - \theta_2^*),\cdots,\mu(\theta_d - \theta_d^*)  \right)$.
> > >
> > > For the first coordinate,we have $\theta_1^{k+1} = \theta_1^k - \eta L(\theta_1^k - \theta_1^*) $, which is equivalent to $\theta_1^{k+1} - \theta_1^* = (1-\eta L)(\theta_1^k - \theta_1^*)$. Similarly, for other coordinates, we have $\theta_i^{k+1} - \theta_i^* = (1-\eta \mu)(\theta_i^k - \theta_i^*), \forall i \in \lbrace2,\cdots,d\rbrace$. To ensure that every step the $\mathbf{\theta}^{k+1}$ closer to $\mathbf{\theta}^*$, we need $0 \leq 1 -\eta L \leq 1$ and  $0 \leq 1 -\eta \mu \leq 1$. Thus, we can set the largest step siza as $\eta = \frac{1}{L}$, which means although we find the optimal solution of the first coordinate in one step, we need at least $\mathcal{O}(L/\mu)$ steps to find the optimal solution of other $d-1$ coordinates. Thus the total communication cost is $\mathcal{O}(dL/\mu)$. If we compress the gradient from dimension $d$ to $1$, even though the estimate is accurate, we still need 1 communication round for the first coordinate and $\mathcal{O}(L/\mu)$ communication rounds for others to find the optimal solution, and the total communication cost is $\mathcal{O}(dL/\mu)$. Though the objective $f$ satisfies "top-heavy" Hessian and bounded trace, the method does not take the advantage of this property. By contrast, SEPARATE introduces common random projection compressor and bound the variance to $\frac{tr(\mathbf{A})}{m}$. This allows us to achieve convergence by communicating the main information. Thus we take the advantage of $tr(\mathbf{A}) \ll dL$, and our algorithm brings a substantial dimension improvement. The results of the previous method only align with the conclusions of our method in the worst case. We think this discussion can solve most of your concerns in Q1.
> > >
> > > ### Q2: Clarification on Equation (46) and Assumption 5.4
> > >
> > >
> > > We show how to get Equation (46) as below. We first take the second-order Taylor expansion as
> > >
> > > $f(\mathbf{\theta}^{k+1}) = f(\mathbf{\theta}^k) + \langle \nabla f(\mathbf{\theta}^k), \mathbf{\theta}^{k+1} -\mathbf{\theta}^k \rangle + \frac{1}{2}\langle \nabla^2 f(\mathbf{\xi})(\mathbf{\theta}^{k+1}-\mathbf{\theta}^k), \mathbf{\theta}^{k+1}-\mathbf{\theta}^k\rangle$,
> > >
> > > where $\mathbf{\xi} = t\mathbf{\theta}^k + (1-t)\mathbf{\theta}^{k+1}, t \in (0,1)$. Then, due to Assumption 5.4, we have $\nabla^2 f(\mathbf{\xi}) \preceq \mathbf{A}$. Thus for all $\mathbf{x} \in R^d$ we have $\mathbf{x}^\top (\mathbf{A}-\nabla^2 f(\mathbf{\xi}))\mathbf{x} \ge 0$, which means $\mathbf{x} ^\top \nabla^2 f(\mathbf{\xi}) \mathbf{x}  \leq \mathbf{x} ^\top \mathbf{A} \mathbf{x}  $. Letting $\mathbf{x}  = \mathbf{\theta}^{k+1}-\mathbf{\theta}^k$ we obtain Equation (46).
> > >
> > > Considering Assumption 5.4, in fact, the upper bound of $\mathbf{A}$ in Assumption 5.4 is $L\cdot \mathbf{I}$. With $\mathbf{A} \preceq L\cdot \mathbf{I}$, we have $tr(\mathbf{A}) \leq dL$. With 'top-heavy" Hessian, in practical application there is $tr(\mathbf{A}) \ll dL$. This leads to the improvement of our algorithm. When $\mathbf{A} = L \mathbf{I}$, it turns to be the worst case of our algorithm and the convergence rate degenerates to $\mathcal{O}(dL/\mu)$ as we show in the response of Q1. However, in most practical applications, especially with "top-heavy" Hessian, there exists a big difference between the two and our algorithm shows improvement.

---

> > > > ### Author Response · Authors · 2024-11-25
> > > >
> > > > ## Part II. Regarding Experimental Details
> > > >
> > > > ### Q3: Regarding the experimental details
> > > > In general, we do not expect our method to show better performance than the baseline by precisely adjusting the hyperparameters like learning rate and batch size as you say. Our method can be viewed more as a plug-and-play component of the existing optimizer with some robustness. Thus,  in our experimental setup, all hyperparameters are **derived from the default settings** of the respective models as trained within their corresponding frameworks.  We do not tune the hyperparameters specifically. Instead, we utilize the **same** configurations to maintain consistency and comparability with established practices. Moreover, our variance analysis and the main theorem demonstrate that, provided the selection of the compression ratio aligns with the conditions in our theoretical analysis (Theorem 5.5 and 5.7), our method can exhibit effective performance. For the reason why SEPARATE performs better than Adam, we acknowledge that this is not a straightforward issue that can be succinctly elucidated under with study.  Therefore, we intend to conduct a more in-depth investigation in future research endeavors.
> > > >
> > > > ## Part III. Comparison with Sketch Method
> > > >
> > > > ### Q4: Add comparison with the sketch method & Q5: Regarding the fast-JL
> > > >
> > > > Sketch has achieved success in matrix reduction and efficient computation. However, the application of such techniques to communication efficient training of LLMs is also a challenge. We appreciate your suggestion to compare with CountSketch method like FetchSGD. We carefully consider the implementation details such as the fused kernel you mentioned to achieve acceleration at the GPU level. Specifically, we implement it directly based on an existing optimizer like Adam. Our implementation utilizes the matrix parallel computation in GPUs environment, the good adaptability of optimizer class in deep learning framework, and realizes the state space reuse based on FetchSGD. Compared to the baseline, the additional computing we've introduced is almost only from CountSketch. However, the single-step wall-clock time is still larger than the baseline as we show below.
> > > >
> > > > | | single-step time|
> > > > |:------|:------|
> > > > | Baseine | 1350ms |       |
> > > > |SEPARATE |1002 ms |
> > > > |FetchSGD |1623 ms |
> > > >
> > > > FetchSGD runs slower than the baseline because the required computation is still heavy compared to the reduced communication overhead, and fast-JL even more so. This also confirms the importance of our "simple but efficient" target in practical applications. Thanks for your valuable insights, we believe that combining fast sketch methods with low communication overhead training is a promising problem for future research.
> > > >
> > > > ---
> > > > We sincerely appreciate your constructive suggestions and believe the discussion, analysis, explanations, and additional experiments significantly improve the quality of our submission. We hope this provides sufficient reasons to raise the score.

---

> > > > > ### Comment · Reviewer_nVYQ · 2024-11-25
> > > > >
> > > > > Dear authors,
> > > > > Thank you for clarifying and addressing my concerns. I have raised my scores and hope the authors could make the contributions and comparisons a bit more explicit in the revision and investigate remaining open questions in future works.

---

> ### Author Response · Authors · 2024-11-24
>
> ## Question & Answer
>
> Thank you for your patience in reading the above part. Then we clarify your concerns one by one.
>
> ### Q1: The novelty of SEPARATE and the theory
> We think the discussions in the **Important Discussion** can answer this part of your concerns. In total, the simplicity of the method is necessary for the acceleration of wall-clock training time. We design such an exquisite and simple method to realize it in training, and the theoretical analysis is also beyond.
>
> ### Q2: The moving average error feedback section
>
>
> Thank you for pointing out the unclear part of our writing in this section. We have refined this part in the update version. You can refer to refined **Section 4.2** for details. To provide a brief overview, we emphasize the utilization of random vectors for projection may engender large deviation, which implies the potential for abrupt fluctuations in the error term $e_n^k$.   This is particularly pronounced under multiple consecutive iterations with a series of random vectors that possess significantly divergent directional properties.   Such a phenomenon can precipitate the entire training process into converging towards an alternative suboptimal region, as graphically illustrated in Figure 3(b) in Section 6.3.
>
> Furthermore, we introduce the moving average error feedback, which, in theory, can regulate the accumulated error to a magnitude on the order of the compression error of the current step.    This technique exhibits commendable performance in practical applications.    Notably, for training from scratch, the absence of error feedback fails to converge, while the absence of moving average error feedback leads to convergence towards an alternative suboptimal region.    These findings show the significance of our method in ensuring the stability and efficacy of the training process.
>
> ### Q3: Experiment and reproducibility concerns
> Thank you for proposing your concerns about the experiments. In fact, we have proposed the detailed experimental setting of our experiments in **Appendix E**, including the details of methods being compared in Table 4 in Appendix E.3,  and the hyperparameter setting of all the methods in Appendix E.1. Specifically, for training from scratch, our global batch size was 8 $\times$ 512. We set the learning rate at 6.0e-4 with the cosine decay down to the minimum 6.0e-5 after 2000 iterations of warm-up. We also used the gradient accumulation and set the gradient accumulation step at 4. We used the global gradient norm clipping of 1 and set the Adam with $\beta_1 =0.9$ and $\beta_2 = 0.95$. For finetuning tasks, we set the data parallelism at 4 and model parallelism at 2 without pipeline parallelism. We set the global batch size at 32, the learning rate of 2e-5, gradient clipping of 2 and gradient accumulation step at 8. To be fair, these hyperparameters are set the same and we follow the default setting in pre-training and fine-tuning corresponding models.
>
> The improvement of error feedback is remarkable in training from scratch as we talk in the response of Q2. Without error feedback, the pre-training is difficult to work. For fine-tuning tasks, the impact of error feedback on the results is small, but still positive. In essence, we did not tune the hyperparameters specifically for SEPARATE; instead, we utilized the same configurations to maintain consistency and comparability with established practices. We have discussed these in detail in updated Appendix A.
>
> ### Q4: What the "top-heavy" eigenvalues have to do with gradient compression
>
> We think the discussion in the second part of **Important Discussion** can answer this question. "Top-heavy" indicates the trace of Hessian is bounded and thus the variance of our estimate is limited and we obtain the main theorem by it.

---

> > ### Author Response · Authors · 2024-11-24
> >
> > ### Q5: The domain of $e$ and the advantages of moving average error feedback
> >
> > Thank you for pointing out our typo in the definition of error. It is not the orthogonal component of the gradient relative to the random projection subspace but the results we have obtained followed.
> >
> > $e_n^k = \arg\min_{e \in \mathbb{R}^d} \frac{\beta}{2}\left\Vert e - \left(\tilde h_n^k -g_n^k\right) \right\Vert^2 +\frac{1-\beta}{2}\left\Vert e-e_n^{k-1} \right\Vert^2 = \left(1-\beta\right)e_n^{k-1} + \beta\left(\tilde h_n^k -g_n^k\right)$
> >
> > The equation above explicitly gives the form of error. The claim “taking moving average of the historical error maintains the stability and variance of accumulated error?” means that using moving average error, we can bound the accumulated error of the top-$k$ iterations up to the error of $k$-th iteration as equation (6).  As a result, the error becomes more stable with lower variance, and the performance is better in pre-training. We have discussed it in the response of Q2, revised Section 4.2, and the proof in Appendix C.
> >
> > ### Q6: Consider other sketching schemes
> >
> > Thank you for proposing this suggestion. We have considered these sketching schemes but they do not work for us. As we show below, in the training task of GPT-2-345M with the same hyperparameter setting, the single-step time cost of our strategy is even smaller than that of fast-JL.
> >
> > | | single-step time|
> > |:------|:------|
> > |SEPARATE |1002 ms |
> > |fast-JL |3324 ms |
> >
> > The main reason is that though these methods (like fast-JL) have lower computational cost in theory, they have more calculation steps in practice. Fast-JL needs to calculate three matrices serially and do multiplication, while our method only needs to generate one random matrix and do multiplication one time. Matrix multiplication can be efficiently parallel computed on the GPU, and serial matrix calculation on the GPU is inefficient in contrast. Thus, we consider to design our method around the "simple but efficient" target.
> >
> > ### Q7: What is the additional cost to communicate common random Gaussian?
> >
> > Thank you for pointing out this problem which we did not explain clearly before. In practice, there is no additional communication cost to generate the common random matrix, because we can set the generator with the same specific seed in each node rather than generating in one node and broadcast. We have discussed it in updated Appendix A.
> >
> > ---
> > We sincerely appreciate your constructive suggestions and believe the discussion, analysis, explanations, and additional experiments significantly improve the quality of our submission. We hope this provides sufficient reasons to raise the score.

---

> ### Author Response · Authors · 2024-11-26
>
> Thank you for you positive feedback and for adjusting the rating. We genuinely appreciate your insightful and constructive comments. We will refine our work according to your comments and investigate remaining problems in future work. Your engagement has greatly enhanced the quality of our work.

---

### Official Review · Reviewer_acpN · 2024-11-04

**Soundness:** 3
**Presentation:** 3
**Contribution:** 3
**Rating:** 6
**Confidence:** 3

**Summary:**

The paper proposes a gradient compression technique called SEPARATE that aims to reduce communication overhead  across multi-device clusters in LLM training. SEPARATE leverages the natural low-rank properties of gradient and Hessian matrices by projecting gradients onto a low-dimensional subspace using common Gaussian random matrices. The accumulated compression errors are then handled by an error-feedback mechanism. This paper shows a 2x speedup in training time for tasks like GPT-2 pre-training and improved performance in fine-tuning LLMs like LLAMA2-7B.

**Strengths:**

- The paper tackles the a very relevant issue of communication overhead in large-scale model training. The writing is clear and well presented. SEPARATE’s design as a simple, plug-and-play gradient compression method makes it highly practical. The paper also presents theoretical proof showing that the convergence rates of SGD and Adam are maintained while using SEPARATE and also shows relevant experiments.

**Weaknesses:**

The method GaLore (Zhao et.al) has proven that gradients are low rank during training. This work can be cited in this context.

Minor:
The experiments compare the training time of SEPARATE with other efficient communication techniques. This is a relevant indicator of communication overhead, but training time also includes computation time which could different for other methods. Therefore reporting bandwidth usage or total data transfer volume in addition to time could help demonstrate  reduction in data exchanged.

**Questions:**

Refer to weaknesses section

---

> ### Author Response · Authors · 2024-11-24
>
> # Response to Reviewer acpN
>
> We sincerely thank the reviewer acpN's approval of our work and the valuable and constructive comments proposed. We have updated the submitted paper and try to clarify your concerns. We try to answer your questions one by one, and hope our clarification could help solve your concerns and improve our submission.
>
> ## Some Related Work
> Thank you for sharing the related work with us, which will undoubtedly assist our readers in finding relevant techniques. We have cited Galore in the new version of our submission. Moreover, it is worth noticing that these methods for memory-efficient training have significant differences from ours. The key challenge we focused on is a simple but effective strategy aimed at reducing communication overhead. Even minor increases in computational cost are undesirable during the frequent communication process. We have further discussed the differences between Galore and ours in updated Appendix A.
>
> ## Report of Data Transfer
> Thank you for your suggestions about reporting the data transfer to demonstrate reduction in data exchanged. we report the model flops utilization (MFU) and throughout (tokens per second) on each GPU when pre-training GPT-2 345M from scratch to prove the data exchanged reduces as below. The results show that we have reduced the data exchanged.
>
> |  | Baseline | SEPARATE|
> |:------ |:------|:------|
> | MFU |3.76% |11.46%|
> | Tokens per second | 379.26 | 501.96 |
>  ---
> We sincerely appreciate your constructive suggestions and believe the discussion and additional experiments significantly improve the quality of our submission. We hope this provides sufficient reasons to raise the score.

---

> > ### Comment · Reviewer_acpN · 2024-11-25
> > **Thank the authors for the feedback**
> >
> > I thank the authors for addressing my questions. I would like to retain my score at this point.

---

> > > ### Author Response · Authors · 2024-11-25
> > >
> > > Thank you for your patient review and constructive comments. These help us improve our submission.

---

### Official Review · Reviewer_Notb · 2024-11-08

**Soundness:** 3
**Presentation:** 2
**Contribution:** 2
**Rating:** 6
**Confidence:** 2

**Summary:**

This paper introduces SEPARATE, a gradient compression technique designed to address communication bottlenecks in large-scale model training. SEPARATE leverages low-rank properties observed in the gradients and Hessians of large language models (LLMs), compressing gradients using random Gaussian projections. The method also includes an error-feedback mechanism to counteract compression-induced errors by averaging historical errors over time, which aims to stabilize training dynamics. Experimental results on models such as GPT-2 and LLAMA2 demonstrate SEPARATE’s potential for up to 2x speedup compared to baselines while maintaining similar accuracy across several downstream tasks.

**Strengths:**

SEPARATE introduces an innovative error-feedback mechanism that effectively addresses the inherent variability associated with random projections by smoothing out compression errors, leading to more stable gradient updates and robust training dynamics. To support its effectiveness, the authors provide a thorough theoretical analysis demonstrating that SEPARATE maintains the convergence rates for non-convex objectives across both SGD and Adam-type optimizers, ensuring that the compression process does not compromise the underlying optimization goals. Additionally, SEPARATE is designed as a flexible, plug-in module that operates independently of specific optimizers or model architectures, making it a versatile tool that can be seamlessly integrated into existing training frameworks. This adaptability simplifies its deployment in diverse training setups, enhancing its practical utility for large-scale distributed model training environments.

**Weaknesses:**

The random Gaussian projection approach employed by SEPARATE, though theoretically sound, introduces variance that is only partially mitigated by its error-feedback mechanism. This variance arises from SEPARATE’s use of fixed Gaussian random matrices in each round, which can sometimes yield suboptimal projections that distort the gradients, affecting convergence stability. The variance analysis indicates that stable convergence relies heavily on precise tuning of the error reset frequency and compression ratio, which compromises SEPARATE’s robustness, particularly in scenarios with limited tuning flexibility due to computational constraints. Moreover, while SEPARATE offers flexibility with varying compression ratios, it lacks a mechanism for dynamically adjusting the compression ratio based on factors like gradient sparsity or model variance. This limitation restricts SEPARATE’s adaptability to evolving model sizes or training dynamics, which may lead to suboptimal performance in cases where manual tuning is impractical. Additionally, the experimental results shown in Table 1 do not provide strong evidence of competitive performance.

**Questions:**

Can SEPARATE generalize to architectures without low-rank gradient properties?
Could a more adaptive error-feedback mechanism improve stability?
Could SEPARATE benefit from dynamically adjustable compression ratios?

---

> ### Author Response · Authors · 2024-11-24
>
> # Response to Reviewer Notb
>
> We sincerely thank the reviewer Notb's for the valuable and constructive comments. We have conducted some experiments to add some adaptability to our method. We try to adaptively set the compression ratio and $\beta$ in moving average error feedback. We propose the preliminary experimental results and hope these could help solve your concerns and improve our submission.
>
> ## Robustness of Our Method
>
> We are grateful for your concerns regarding the robustness of our method and would like to address them by clarifying three key aspects. First, our variance analysis and the main theorem demonstrate that, provided the selection of the compression ratio aligns with the conditions in our theoretical analysis (Theorem 5.5 and 5.7), our method can exhibit effective performance. This theoretical underpinning ensures that our method is robust within the specified compression ratio. Second, as a result, in practical applications, especially for models with a substantial number of parameters, such as those in the millions or billions, we can defaultly set the compression ratio as 16, 32, or 64. As illustrated in Table 3, the performance differences between compression ratios of 16, 32, or 64 times for gradient information are minimal. Consequently, the choice of compression ratio is more influenced by the user's device constraints rather than the intrinsic characteristics of the model. Third, in our experimental setup, all hyperparameters are derived from the default settings of the respective models as trained within their corresponding frameworks. We do not tune the hyperparameters specifically for SEPARATE. Instead, we utilize the same configurations to maintain consistency and comparability with established practices. We have discussed these in detail in updated Appendix A.
>
>
>
> ## General Version of SEPARATE
>
> Thank you for the interesting suggestion on adaptive extension of our method. We have carefully considered **your requirements** and extended our algorithm with adaptive compression ratio and $\beta$. We show the algorithm as below. You can refer to the formal version in updated **Appendix E.5**.
>
>
> **Algorithm:** G-SEPARATE: General Version of Simple Low-rank Projection
>
> **Input:**
> - Initialization model parameters with $L$ layers, $N$ nodes, layer-wise communication ratio $\lbrace m_l \rbrace _ {l=1}^{L}$,  layer-wise $\lbrace \beta_l\rbrace_{l=1}^{L}$, $\beta_l=0.95$, $\forall l \in [L]$, error reset frequency $T_e$, adaptive update frequency $T_a$, a common Gaussian random number generator, initialize $e^0 \in \mathcal{B}(\mathbf{0},c_1)$
>
> **While** $k \leq K$ **do:**
>
> 1. **In each node $n$ compute stochastic gradient $g_{n,l}^k$ and $h_{n,l}^k = g_{n,l}^k + e_{n,l}^k$;**
> 2. **Generate fresh i.i.d. common random Gaussian vectors $\xi_1,\cdots,\xi_{m_l^k}\sim N(0,I_d)$ and compute $[p_{1,n,l},\cdots,p_{m_l^k,n,l}]$ with $p_{i,n,l} = \langle h_{n,l}^k,\xi_i\rangle$ as the low-dimension projection of $h_{n,l}^k$;**
> 3. **Do all-reduce and obtain global projected gradient $[\tilde p_{1,l},\cdots,\tilde p_{m_l^k,l}]$;**
> 4. **Compute $\tilde h_{n,l}^k = \frac{1}{m_l^k} \sum_{i=1}^{m_l^k} \tilde p_{i,l} \cdot \xi_i$ and use $\tilde h_{n,l}^k$ for model weight update in node $n$;**
> 5. **Update error:**
>    - $e_{n,l}^{k+1} = (1-\beta_l^k)e_{n,l}^k + \beta_l^k (h_{n,l}^k -\tilde h_{n,l}^k)$ if $k$ % $T_e \not= 0$;
>    - $e_{n,l}^{k+1} = 0$ if $k$ % $T_e =0$ (error reset);
> 6. **Layer-wisely update compression ratio and $\beta_l$:**
>    - $m_l^{k+1} = {\rm int}\left(1 + m_l^k \cdot \left(1 + \frac{\langle \tilde h_{n,l}^k, g_{n,l}^k \rangle}{\Vert \tilde h_{n,l}^k \Vert \cdot \Vert g_{n,l}^k \Vert} \right)\right),$
>    - $\beta _ l^{k+1} = \max\left\lbrace\min\left\lbrace\beta_l^k \cdot\left(1 + \frac{\langle \tilde h_{n,l}^k, g_{n,l}^k \rangle}{\Vert \tilde h_{n,l}^k \Vert \cdot \Vert g_{n,l}^k \Vert} \right), 0.99 \right\rbrace, 0.90 \right\rbrace$ if $k$ % $T_a = 0$;
>    - $m_l^{k+1} = m_l^k$,
>    - $\beta_l^{k+1} = \beta_l^k$, if $k$ % $T_a \not= 0$;
>
> **End While**

---

> > ### Author Response · Authors · 2024-11-24
> >
> > We expect to dynamically adjust these hyperparameters by combining the characteristics of the parameters at each layer of the model and the training dynamic. The adaptive strategy needs to remain simple and efficient to ensure the overall training wall-clock time is reduced. Thus, we still do not consider the strategies that require periodic heavy computations, such as periodic SVD. Under this consideration, we get G-SEPARATE.
> >
> > We pre-train GPT-2-345M on 10B tokens OpenWebtext dataset from scratch to verify the effectiveness of G-SEPARATE. We follow the same hyperparameter setting of our pre-training experiment in Appendix E.1. We set the adaptive update frequency $T_a = 2000$ to ensure the stability of training. The results shown in Figure 4 indicate that the general version more slightly fits the baseline, but shares a similar performance to the original version in total.
> >
> > ---
> > We sincerely appreciate your constructive suggestions and believe the discussion and additional experiments significantly improve the quality of our submission. We hope this provides sufficient reasons to raise the score.

---

> > > ### Comment · Reviewer_Notb · 2024-11-26
> > >
> > > Most of my concerns have been addressed. I am adjusting my rating accordingly.

---

> > > > ### Author Response · Authors · 2024-11-26
> > > >
> > > > Thank you for you positive feedback and for adjusting the rating. We genuinely appreciate your insightful and constructive comments. Your engagement has greatly enhanced the quality of our work.

---

### Official Review · Reviewer_Kod5 · 2024-11-08

**Soundness:** 3
**Presentation:** 3
**Contribution:** 3
**Rating:** 6
**Confidence:** 3

**Summary:**

This work proposes an efficient way to communicate gradients between nodes for training large models. The method works by communicating random projections of the gradients, where the projections have zero mean and identity covariance. The projections can be regenerated from the random seed, so a shared random number generator between nodes is assumed. Thus, since the projections $P$ have identity covariance, an unbiased estimate of the original gradient $g$ can be found by noting $\mathbb{E}[gPP^\top]=g$. The authors further improve this method by incorporating an error-feedback mechanism that works like momentum. They derive an Adam variant of this method and show that it is effective in practice. They also provide convergence analysis of this method on non-convex objectives.

**Strengths:**

- Using the random seed for reconstructing the random projections is a smart trick for efficient communication and saving space.
- This method has the benefit of being an unbiased estimator of the true gradient, which is not the case with low-bit and some low-rank methods. Even though the variance might be bad (i.e., worse than the distance between the true gradient and its truncated SVD), error feedback (with restarts) seems to mitigate this issue during training, allowing SEPARATE to be stable and perform better than low-rank projection methods, such as PowerSGD. This implies that old gradient directions can still be relevant even after many iterations.
- The authors additionally show an Adam variant of SEPARATE and derive the convergence rate of both the normal variant and the Adam variant.
- The low-rankedness in LLMs have been (empirically) demonstrated before, but the authors here show a theoretical analysis of this phenomenon in the appendix.

**Weaknesses:**

- Sharing random generator between nodes might introduce a layer of complexity that could potentially make debugging difficult when things go wrong. For example, this might prevent a real-life implementation to use non-deterministic approaches for accelerating training. Elaborating on this part would be great. For example, the authors can emphasize that this is not a problem and discuss how to ensure a proper regeneration of the random variable across nodes.
- This trick for getting an unbiased estimator from a sketch matrix M is actually not new as far as I'm concerned. The authors did not claim its novelty, but I just wanted to mention this since they did not cite relevant works to this trick. For example, the Hutchinson estimator does a very similar thing to calculate the trace (or diagonal) of the Hessian. Citing this estimator can help future readers to look for relevant tricks in the literature.
- Table 1: the authors should explain what "performance" mean here, at least in the main text. I also do not think the use of average performance across datasets is sound. Perhaps the average rank or quantile per dataset is a better aggregated metric.
- Regarding regularization by projection, an elaboration of the claim in line 407 would be great. If the authors believe that some sort of regularization is happening, then it would be great to provide some experiments or analysis to corroborate this hypothesis, which would be interesting to see. Otherwise, this might feel like a handwavy explanation of the results.
- The discussion regarding the hessian spectrum is already known and have been demonstrated in quite a few works, e.g. [1], which the authors duly noted, but I'm mentioning to emphasize that this is a motivation rather than a novel discovery. This also applies to studies regarding low-rank properties in LLMs. Thus, the originality of the contribution comes from applying random low-rank projections with an error-feedback mechanism along with the analysis in the appendix. Perhaps stating this clearly in the contributions section can help understand this work better.
- Some explanations are provided without support, such as line 458: “When the random projection directions are far from the dominant directions of Hessian in several continuous iterations, the variance of error will become extremely large and misguide the next iteration.” It is easy to see how this holds intuitively, but it might not necessarily be the case that error feedback mitigates this phenomenon, or that this phenomenon occurs in the first place in practice. For example, SEPARATE1 in Table 2 seems to be working fine.
- I don't see how the Ablation study in Table 2 is conclusive. It would be better to provide the average + std for a few seeds, say 5. That would make it clearer whether the difference is significant or not.

[1] Empirical Analysis of the Hessian of Over-Parametrized Neural Networks. Sagun et al. ICLR 2018

**Questions:**

None.

---

> ### Author Response · Authors · 2024-11-24
>
> # Response to Reviewer Kod5
>
> We sincerely thank the reviewer Kod5's approval to our work and the valuable and constructive comments proposed. We have updated the submitted paper and try to clarify your concerns. We try to answer your questions one by one, and hope our clarification could help solve your concerns and improve our submission.
>
> ## Question & Answer
>
> ### Q1: Sharing random generator between nodes might introduce a layer of complexity
>
> Thank you for pointing out this matter and for your suggestions. We appreciate your feedback on areas where we could improve our explanations and clarifications. First, in practice, there is no additional communication cost involved, because we can initialize the random variable generator with **the same specific seed** on each node, rather than generating it in one node and then broadcasting it to others. This ensures efficiency and avoids extra communication overhead. We have discuss this in updated Appendix A. Second, we have conducted an assessment of the introduced randomness with the aim of evaluating the influence of random seeds on our results, as detailed in Section 6.3. A more comprehensive discussion on this topic is provided in the response to Q7.
>
>
> ### Q2: About previous estimator and ours
>
> Thank you for sharing the related work with us, which will undoubtedly assist our readers in finding relevant techniques. We have incorporated this related work into the new version of our submission in line 143 and 144. Moreover, we have provided a clarification of our approach compared with other previous estimator for memory-efficient training. The key challenge we focused on is a **simple but effective** strategy aimed at reducing communication overhead. **Even minor increases in computational cost are undesirable during the frequent communication process**, as we have detailed shown in Introduction and updated Appendix A. To achieve this, we have developed an improved moving average error feedback technique, which is particularly effective when training from scratch. Our strategy is independent of the optimizers and can be seamlessly integrated into FSDP. In contrast, previous methods were either challenging to implement or were limited to DDP applications. We have discussed these in updated Appendix A.
>
>
>
> ### Q3: Some concerns of Table 1
>
> Thank you for bringing this issue to our attention, which we acknowledge was not previously explained with sufficient clarity.  We have taken your feedback and have provided a detailed explanation of this in line 413 and 414 in the updated version of our submission. To clarify, when we refer to "performance," we are indicating the scores achieved on the relevant benchmarks.  The practice of taking the average score is a widely accepted method for evaluating LLMs [1,2].  In line with your suggestion, we have also taken into account the average rank and have included this metric in Table 1 for a comprehensive assessment. We show the results below.
>
> | Methods | Average Rank |
> | :------ | :------|
> | Adam | 3.22 |
> | PowerSGD | 3.00 |
> | 1-bit Adam | 3.22 |
> | ZeRO++ | 3.22 |
> | SEPARATE | **2.22** |
>
>
>
> ### Q4: Regarding regularization by projection
> We appreciate your observation regarding the limitations in our previous discussion. It is important to note that our initial remarks constituted a preliminary hypothesis and interpretation of the observed phenomenon. We acknowledge that this is not a straightforward issue that can be succinctly elucidated under with study; therefore, we intend to conduct a more in-depth investigation in future research endeavors. Consequently, in the current version of our work, we have elected to delete the aforementioned sentence to avoid any potential misinterpretation or oversimplification of the complex problem at hand.

---

> ### Author Response · Authors · 2024-11-24
>
> ### Q5: Motivation V.S. new discovery
>
> We extend our sincere gratitude for highlighting the ambiguity present in our submission. We have meticulously refined this part within the updated version, with a clear distinction between the motivation and our contributions in the **Introduction**. We underscore that the observation of the "top-heavy" Hessian is the cornerstone of our research motivation, and we present both theoretical and empirical evidence of this phenomenon in Section 3. Subsequently, we propose our contributions about designing an efficient algorithm for communication-efficient training, which encompasses the common random projection technique, an improved moving average error feedback mechanism, theoretical analyses, and corroborative experimental results.
>
>
> ### Q6: Some explanations and support
>
> We appreciate your suggestions regarding the areas in our previous submission that required further elucidation. We have revised the description of error feedback in the updated version. First, we would like to emphasize a critical aspect of single-step iteration within our method. Specifically, the use of random vectors for projection may introduce significant deviation (although this doesn't always happen). This implies the potential for abrupt fluctuations in the error term $e_n^k$, particularly under consecutive iterations with a sequence of random vectors that exhibit markedly different directions. Such fluctuations can precipitate the entire training process into converging towards an alternative suboptimal region, as shown in **Figure 3(b)** in Section 6.3. In practical applications, we have noted a significant improvement in the training process when moving average error feedback is incorporated, especially when training from scratch. Referred to Section 6.3, the absence of error feedback renders the pre-training phase challenging to execute effectively.
>
>
>
> ### Q7: Concerns of ablation study in Table 2
>
> Thank you for providing this suggestion for the report on the ablation study. What we want to emphasize is that for the baseline, our method and other methods to be compared, we use the greedy sample for the generation process. The results fluctuate less so we do not report as average + std form in the original version. We take your suggestion into consideration and change the sampling method, but it takes time to modify all the evaluation experiments due to equipment constraints. We report the results of the experiments in Table 2 that have been carried out below, and we will update it in the coming days. As stated below, the difference is small enough.
>
> |  | GSM8K | MBPP | NQ | Arc-e | Arc-c | PIQA |
> |:------| :------|:------|:------|:------|:------|:------|
> | SEPARATE5 | 19.53 $\pm$ 0.64 |21.20 $\pm$ 0.20 |4.15 $\pm$ 0.03 | 73.37 $\pm$ 0.70 |50.34 $\pm$ 0.51 |52.91 $\pm$ 0.08 |
>
>
> ---
> We sincerely appreciate your constructive suggestions and believe the discussion, analysis, explanations and additional experiments significantly improve the quality of our submission. We hope this provides sufficient reasons to raise the score.
>
> [1] Edward Beeching et al. Open LLM Leaderboard (2023-2024).
>
> [2] Abhimanyu Dubey et al. The Llama 3 Herd of Models.

---

> > ### Comment · Reviewer_Kod5 · 2024-11-25
> >
> > Thank you for the detailed rebuttal. I am satisfied with the response and would like to keep my current rating.

---

> > > ### Author Response · Authors · 2024-11-26
> > >
> > > Thank you for you positive feedback. We genuinely appreciate your insightful and constructive comments. Your engagement has greatly enhanced the quality of our work.

---

### Meta-Review · Area_Chair_1Z8h · 2024-12-19

**Metareview:**

This paper focuses on solving communication problems in the distributed training of Large Language Models (LLMs). The authors propose a method called SEPARATE, which uses low-rank properties of gradients and Hessians to compress gradients. The method uses random Gaussian projections and an error-feedback mechanism to reduce communication costs. The authors provide a theoretical analysis of SEPARATE. The experiments show that SEPARATE speeds up training by up to 2× for GPT-2-Medium and works well for fine-tuning LLAMA2-7B models.

The main strengths of the paper are its simple design, easy integration with existing optimizers, and good performance in practice. The authors clearly show how SEPARATE improves training speed while keeping accuracy high. During the review process, the authors addressed concerns about the novelty of the method compared to similar works like GaLore and Flora. They also improved their explanation of the error-feedback mechanism. There are some minor weaknesses, such as the need for careful tuning of hyperparameters and the possible variance caused by random projections. However, the method is still robust and effective.

**Additional Comments On Reviewer Discussion:**

The reviewers had a productive discussion regarding the strengths and weaknesses of the paper. Initially, there were concerns about the novelty of the proposed method, especially in comparison to similar techniques like GaLore and Flora. However, the authors clarified how SEPARATE addresses communication bottlenecks specifically, which is different from methods focused on memory efficiency. The reviewers also questioned the robustness of the error-feedback mechanism and the potential variance introduced by

---

### Decision · Program_Chairs · 2025-01-22

Accept (Poster)